cellular biology/health and disease and epidemiology

geranylgeraniol, mesenchymal stem cell, zoledronic acid, cytotoxicity, RhoA, YAP

**Author for correspondence:**
Weerachai Singhatanadgit
e-mail: s-wrch@tu.ac.th

†Present address: Faculty of Dentistry, Thammasat University (Rangsit campus), 99 Moo 18 Pahonyothin Road, Klong Luang, Pathumthani, 12120, Thailand.

# Geranylgeraniol prevents zoledronic acid-mediated reduction of viable mesenchymal stem cells via induction of Rho-dependent YAP activation

Weerachai Singhatanadgit[1,2,†], Weerawan Hankamolsiri[3] and Wanida Janvikul[3]

[1]Faculty of Dentistry, and [2]Research Unit in Mineralized Tissue Reconstruction, Thammasat University, Pathumthani, 12121, Thailand
[3]Biofunctional Materials and Devices Research Group, National Metal and Materials Technology Center, Pathumthani 12120, Thailand

WS, 0000-0001-7594-1305

Long-term use of zoledronic acid (ZA) increases the risk of medication-related osteonecrosis of the jaw (MRONJ). This may be attributed to ZA-mediated reduction of viable mesenchymal stem cells (MSCs). ZA inhibits protein geranylgeranylation, thus suppressing cell viability and proliferation. Geranylgeraniol (GGOH), which is a naturally found intermediate compound in the mevalonate pathway, has positive effects against ZA. However, precise mechanisms by which GGOH may help preserve stem cell viability against ZA are not fully understood. The objective of this study was to investigate the cytoprotective mechanisms of GGOH against ZA. The results showed that while ZA dramatically decreased the number of viable MSCs, GGOH prevented this negative effect. GGOH-rescued ZA-exposed MSCs formed mineralization comparable to that produced by normal MSCs. Mechanistically, GGOH preserved the number of viable MSCs by its reversal of ZA-mediated Ki67+ MSC number reduction, cell cycle arrest and apoptosis. Moreover, GGOH prevented ZA-suppressed RhoA activity and YAP activation. The results also established the involvement of Rho-dependent YAP and YAP-mediated CDK6 in the cytoprotective ability of GGOH against ZA. In conclusion, GGOH preserves a pool of viable MSCs with osteogenic potency against ZA by rescuing the activity of Rho-dependent

YAP activation, suggesting GGOH as a promising agent and YAP as a potential therapeutic target for MRONJ.

## 1. Introduction

In patients receiving antiresorptive drugs, prevention and treatment of complications, including medication-related osteonecrosis of the jaw (MRONJ) [1], have become major concerns in current dental practice, especially involved in the management of osteonecrosis of alveolar socket bone after tooth extraction. Nitrogen-containing bisphosphonates (N-BPs) are a group of potent antiresorptive drugs that mainly inhibit osteoclast formation and functions. N-BPs target bone due to their high affinity for hydroxyapatitite, particularly in areas of high bone remodelling, and the bound N-BPs are released by osteoclasts, their main cellular target [2]. However, emerging data on the development of MRONJ raise new concerns about the duration and cumulative dose of N-BP administration. The cumulative incidence of developing MRONJ among patients receiving intravenous N-BPs increased from 0.5–0.8% at 1 year to 1.3–4.3% at 3 years without a plateau after 2–3 years as reported for patients receiving an alternative antiresorptive drug denosumab [1,3]. The increasingly accumulated N-BPs in the long-term use of N-BPs may be expected in the bone of patients, and the high level released during bone remodelling is considered to increase the risk of developing MRONJ [4–6]. It has been shown that in addition to its inhibition of osteoclast activity, high-dose zoledronic acid (ZA), the most potent N-BP highly relevant to MRONJ, decreased numbers of osteoblasts in both *in vivo* and *in vitro* experiments [7]. Our previous *in vitro* study further supports the inhibitory effect on osteoblastic lineage cells by showing that while low-dose ZA inhibited osteoblast differentiation of mesenchymal stem cells (MSCs), high-dose ZA suppressed the viability of MSCs [8]. These findings support the involvement of high-dose N-BP cytotoxicity to osteoblast lineage cells in MRONJ. Thus far, there is a lack of evidence to support effective prevention and treatment of MRONJ due to N-BPs [9], prompting the investigation to better understand their molecular targets in osteogenic cells beyond osteoclasts and identify key molecules that help protect cells against N-BP cytotoxicity.

Although not yet fully understood, N-BPs inhibit farnesyl diphosphate synthase, a key enzyme of the intracellular mevalonate pathway, resulting in a reduction of isoprenoid intermediates, such as geranylgeranyl pyrophosphate (GGPP), and in turn altering protein geranylgeranylation which is required for the post-translational maturation of the small GTP-binding proteins [10]. The Rho GTPase family, which is one of the major branches of the Ras superfamily of small GTPases, is subdivided into several subfamilies, including the Rho (isoforms A, B and C), Rac (isoforms 1, 2, 3) and Cdc42 subfamilies. The Rho subfamily GTPases play an important role in the control of cell cycle and cell proliferation [11]. High RhoA GTPase activity maintains the undifferentiated mesenchymal cell phenotype [12]. The inhibition of these Rho subfamily GTPases, induced by malfunctioning protein geranylgeranylation, not only suppresses the formation and function of osteoclasts, but also interferes with the growth, differentiation and function of osteoblast lineage cells, resulting in impaired osteogenesis [13].

Geranylgeraniol (GGOH) is a naturally found intermediate compound in the mevalonate pathway essential for cell viability and proliferation [14]. It has positive effects against N-BPs in many cell types by facilitating protein geranylgeranylation, improving cell viability and proliferation in tissue regeneration, thus overcoming N-BP-induced apoptosis [15,16]. Some studies had supported the use of GGOH in angiogenesis therapy [17] and local toxicity therapy [18]. The cytotoxic effect of ZA on human osteoblasts and its reversal by GGOH suggested that the effect of ZA on this cell type was mediated via the mevalonate pathway [13]. The cytotoxic effect of ZA on a cancer cell line has been shown to be reversed by GGOH-rescued RhoA activity [19].

Yes-associated protein 1 (YAP1 also known as YAP) is a transcriptional regulator that possesses significant biological functions in tissue development and homeostasis. YAP activity is critical for whole organ growth, for the expansion of progenitor cells during tissue regeneration, and for cell proliferation [20]. It has been shown that YAP promotes proliferation, impedes apoptosis and delays the senescence of stem cells derived from human periodontal ligament cells [21]. A previous study further suggested a critical role of YAP in maintaining stem cell pluripotency [22]. Phosphorylation of YAP at Serine 127 resulted in cytoplasmic accumulation of YAP and inhibited its activity in the nucleus [20]. However, when activated, dephosphorylated cytoplasmic YAP molecules translocated into the nucleus, activated its target genes and elicited biological functions, the processes of which are tightly regulated [23,24]. It has been shown that the GGPP produced by the mevalonate cascade is

required for the activity of Rho GTPases that, in turn, regulates the activation of YAP by inhibiting their phosphorylation and promoting their nuclear accumulation [25]. However, whether or not the exogenously added GGOH can reverse these inhibitory effects of ZA on the functions of Rho and YAP, and thus MSC viability, is not yet known. Therefore, the present study aimed to investigate the cytoprotective mechanisms of GGOH against ZA in MSCs. The present study showed that ZA suppressed MSC viability via inhibiting Rho-induced YAP activation which was rescued by GGOH-facilitated activation of Rho-YAP signalling, thus preserving a pool of viable MSCs that retained their osteogenic potency.

# 2. Material and methods

## 2.1. Cell culture

In the present study, human MSCs (Lonza Biologics plc, Cambridge, UK) were maintained in a standard medium consisting of the α-minimum essential medium (α-MEM) (Gibco Life Technologies Ltd, Paisley, UK) containing 15% fetal calf serum (FCS) (PAA Laboratories, Yeovil, UK) supplemented with 200 U ml$^{-1}$ penicillin, 200 µg ml$^{-1}$ streptomycin and 2 mM L-glutamine (all from Gibco) at 37°C in a humidified atmosphere of 5% $CO_2$ in air. The media were changed every 2–3 days. Cells between passages 4–6 were used in the study.

For osteogenic differentiation induction, MSCs were seeded at a density of $1.5 \times 10^4$ cells cm$^{-2}$ and allowed to grow with the standard medium for the first 48 h until the cells reached 80% confluence. Then, the cells were incubated with an osteogenic medium (OM) (standard medium with 100 nM dexamethasone, 50 µM ascorbate-phosphate and 10 mM β-glycerolphosphate) (all from Sigma-Aldrich, St Louis, MO, USA).

## 2.2. Treatment of cells

For cell viability study, MSCs were cultured with ZA (Novartis Pharmaceuticals, UK) (0, 5 or 50 µM) and GGOH (Sigma-Aldrich) (0–100 µM) for 1–7 days. To inhibit the Rho activity, cells were pretreated with 30 µM Rhosin hydrochloride (Rhosin; Tocris Bioscience, Minneapolis, MN) 30 min prior to the addition of ZA and GGOH, and after 24 h, cells were immunostained for YAP. Lysophosphatidic acid (LPA) and dobutamine hydrochloride (DH) (both from Sigma-Aldrich, St Louis, MO, USA) were used to activate and suppress YAP in MSCs, respectively. Cells were treated with 20 µM DH or 10 µM LPA in addition to ZA and GGOH treatments, and after 3 days, cells were subjected to cell viability assay. In some experiments, MSCs were treated with palbociclib (1 µM) (Sigma-Aldrich) to inhibit cyclin-dependent kinases CDK4/6 in addition to ZA and GGOH treatments for 3 days in culture, and the cell viability was determined.

## 2.3. Morphological analysis of MSCs and cell viability assay

Cells were cultured at a density of $3 \times 10^3$ cells well$^{-1}$ in 96-well plates for 18 h. Cells were treated as indicated in each experiment. After indicated time in culture, the cells were fixed with 4% paraformaldehyde (PFA) for 15 min and stained with 0.05% w/v crystal violet solution for 15 min. The cell morphology was examined under a light microscope (Nikon Eclipse TS100), and photomicrographs of cell appearances were taken using a Nikon Digital sight DS-L2.

For cell viability assay, 3-(4,5-dimethylthiazol-2-yl)-2,5-diphenyltetrazolium bromide (MTT) assay was used to examine the viability of MSCs. After being cultured at the indicated times, the cells were incubated with 0.2% MTT solution for 4 h at 37°C, and the reaction was then stopped by adding dimethyl sulfoxide (DMSO) and glycine buffer. The end product colour was subsequently analysed by measuring an absorbance at 490 nm ($A_{490}$) which corresponds to the viability of cells. Cell viability is expressed as the mean percentage of control (100%). Data are presented as the mean percentage ± s.d. from three independent experiments.

## 2.4. Quantitative real-time PCR (Q-PCR)

Q-PCR was performed to examine the expression of genes associated with osteoblast differentiation, i.e. runt-related transcription factor 2 (RUNX2), type-I collagen (COL-I), alkaline phosphatase (ALP) and osteocalcin

(OCN), and cell proliferation, i.e. cyclin-dependent kinase 6 (CDK6). Total RNA was isolated and first-strand cDNA was synthesized from 1 µg RNA. The first-strand cDNA was subjected to Q-PCR using SYBR Green I dye performed in an iQ5 iCycler (BioRAd, Bradford, UK), with specific primers for the RUNX2, COL-I, ALP, OCN, CDK6 and GAPDH mRNA. GAPDH was used as an endogenous control. SYBR Green PCR reaction mixtures using SYBR Green I Master kit (Roche Diagnostic Co.) were set up as suggested by the manufacturer. The amplification conditions consisted of 40 cycles at 95°C for 15 s, followed by 60°C for 30 s and subsequently 72°C for 30 s. The specificity of the PCR products was verified by melting curve analysis. The PCR reactions were performed in six replicates, and each of the gene signals was normalized to the GAPDH signal in the same reaction. The mRNA expression is expressed as the mean fold-change of control (1.0). Data are presented as the mean fold-change ± s.d. from three independent experiments. Three biological replicates were used to assess the biological variability of the gene expression results, and similar gene expression patterns were obtained from all biological replicates. Primer sequences were as follows: RUNX2 F 5′-TGGTTACTGTCATGGCGGGTA-3′, R 5′-TCTCA GATCGTTGAACCTTGCTA-3′; COL-I F 5′-GAGGGCCAAGACGAAGACATC-3′, R 5′-CAGATCACGTCA TCGCACAAC-3′; ALP F 5′-ACTGGTACTCAGACAACGAGAT-3′, R 5′-ACGTCAATGTCCCTGATGTTATG-3′; OCN F 5′-CACTCCTCGCCCTATTGGC-3′, R 5′-CCCTCCTGCTTGGAC ACAAAG-3′; CDK6 F 5′-GCTGACCAGCAGTACGAATG-3′, R 5′- GCACACATCAAACAACCTGACC-3′; GAPDH F 5′-CTGGGCTACACTGAGCACC-3′, R 5′- AAGTGGTCGTTGAGGGCAATG-3′ [26,27].

## 2.5. Biomineralization assay

Biomineralization was determined by an alizarin red S staining assay. Under the standard medium or OM for 14–21 days, cells were fixed with cold methanol for 30 min at 4°C and washed with distilled water. Subsequently, 1% alizarin red S (pH 4.2; Sigma) was added onto the samples and incubated for 10 min at room temperature, and the samples were rinsed twice with methanol to remove unbound alizarin red S. Alizarin red S-stained mineralized matrices observed as bright red deposits were photographed under a Nikon digital camera.

## 2.6. Analysis of Ki67$^+$ proliferating cells, cell cycle and cell apoptosis by flow cytometry

Flow cytometric analysis was performed using the CytoFLEX Flow Cytometer and the CytExpert software (both from Beckman Coulter, CA, USA) for data acquisition and analysis, respectively. For analysis of Ki67$^+$ cells, cells were stained with anti-human Ki67 antibody conjugated with FITC (Miltenyi Biotech, CA, USA), by following the manufacturer's instructions. Data are presented as the mean percentage of Ki67$^+$ cells ± s.d. from three independent experiments. For cell cycle analysis, cells were collected, and the cell pellets were resuspended in ice cold 70% ethanol for 30 min. Cells were then rinsed twice in PBS and resuspended in 20 µg ml$^{-1}$ propidium iodide (PI) in PBS with 50 µg ml$^{-1}$ RNase A (both from Sigma) at 4°C for 30 min and subjected to flow cytometry. The distribution of cells in three major phases of the cycle (G1 versus S versus G2/M) was analysed and the data are presented as the mean percentage ± s.d. from three independent experiments. Cell apoptosis was measured by staining the cells with annexin V-FITC and PI (Miltenyi Biotech), by following the manufacturer's instructions, and the stained cells were analysed by flow cytometry. The annexin V$^+$PI$^-$ cells were considered to be apoptotic cells. Cell apoptosis is expressed as the mean percentage of the total cells tested, and the data are presented as the mean percentage ± s.d. from three independent experiments.

## 2.7. Confocal immunofluorescence of YAP

For immunolocalizing YAP, cells were cultured at a density of $3 \times 10^3$ cells cm$^{-2}$ in an eight-well chamber slide (Sigma) for 18 h. After indicated treatments for 24 h, cells were fixed with 4% paraformaldehyde in PBS for 15 min and rinsed three times in PBS for 5 min each. Cells were blocked with blocking buffer (5% normal goat serum and 0.3% Triton™ X-100 in PBS) for 60 min and incubated in a rabbit monoclonal anti-YAP antibody (#14074, Cell Signalling) diluted 1 : 200 in dilution buffer (1% BSA and 0.3% Triton X-100 in PBS) overnight at 4°C. The samples were then incubated with Alexa Fluor 488-conjugated goat anti-rabbit IgG antibody (Invitrogen, CA, USA), diluted 1 : 2000 in dilution buffer for 2 h at room temperature in the dark and followed by nuclear staining 1 µM DAPI (Sigma). Finally, samples were mounted with mounting media (DAKO), sealed with nail polish and investigated under a confocal fluorescence microscope (Nikon Ti Eclipse, Nikon Instruments Inc., NY, USA). The NIS-Elements

software was used for the image analysis, with Z-stacks of images being acquired for each channel. The middle confocal slice was chosen from the images at the middle nuclear plane, detected in the DAPI channel, and on the same slice, the image was acquired in the YAP channel to determine the localization of YAP. Nuclear localization of YAP was determined by the presence of co-localization of YAP and DAPI with the intensity ratio of nuclear localization to cytosolic localization being higher than 1.0. The percentage of nuclear YAP-positive cells was measured from a total of 100 cells. Data are presented as the mean percentage ± s.d. from three independent experiments.

## 2.8. Detection of GTP-bound RhoA

The detection of GTP-bound RhoA (RhoA-GTP) was used to determine the activity of RhoA. GTP-bound RhoA was measured from cell lysates using a RhoA activation G-LISA kit (BK124, Cytoskeleton, Inc., CO, USA), by following the manufacturer's instructions. Briefly, equal numbers of control and treated cells were lysed at the indicated time points using the provided cell lysis buffer and a protease inhibitor cocktail (Thermo Fisher Scientific). Lysates were quantified using the BCA protein assay, and aliquots of 50 μg of protein from each sample were loaded onto the G-LISA plate for GTP-bound RhoA analysis. Results were obtained by measuring $A_{490}$ which corresponds to the activity of RhoA. RhoA activity is expressed as the mean percentage of control (100%), and the data are presented as the mean percentage ± s.d. from three independent experiments.

## 2.9. siRNA transfection

MSCs were seeded at $6 \times 10^3$ cells cm$^{-2}$ in an antibiotic-free basal medium 24 h prior to transfection. siRNAs and transfection reagents were purchased from Thermo Fisher Scientific (USA), and the transfection was performed following the manufacturer's protocol. Briefly, Silencer® Select human YAP1 siRNA (s20368) and Silencer® Select Negative Control No. 1 siRNA at a final siRNA concentration of 50 nM were transfected using Lipofectamine™ RNAiMAX in Opti-MEM® I Reduced Serum Medium for 6 h. After 6 h, the medium was replenished with the antibiotic-containing standard medium. After 48 h, the transfected cells were collected for Western blot analysis of YAP and subsequent functional assays. Cells treated with Lipofectamine™ RNAiMAX only (non-transfected control) and cells transfected with a Silencer negative control siRNA (control siRNA) were used as experimental controls.

## 2.10. Western blot analysis

Total protein was extracted from cells using a RIPA cell lysis buffer containing protease and phosphatase inhibitor cocktails (Thermo Fisher Scientific). The total amount of protein was quantified in RIPA extracts using the BCA kit. Equivalent quantities of RIPA-solubilized proteins were resolved by 7–14% SDS/polyacrylamide gels, and the separated proteins were transferred to nitrocellulose membranes (Merck Millipore, MA, USA). Membranes were blocked with 5% non-fat milk in Tris Buffered Saline with 0.1% Tween® 20 (TBST) and probed with the following primary antibodies: anti-YAP, anti-phospho-YAP Ser127 (pYAP), anti-CDK4, anti-CDK6 (all from Cell Signaling Technology) and anti-α-tubulin (Santa Cruz) diluted 1 : 1000 in 5% BSA in TBST overnight at 4°C. Primary antibody-probed blots were visualized with anti-rabbit IgG-horseradish peroxidase (Santa Cruz) diluted 1 : 5000 in 5% BSA in TBST for 1 h at room temperature. Protein detection was performed by autoradiography using enhanced chemoluminescence (Merck Millipore). Films were scanned and saved as digital images, and the intensities of protein bands were analysed using ImageJ software (National Institutes of Health, MD, USA). The relative level of protein, normalized by the loading control α-tubulin level, is expressed as mean fold-change of control (1.0) ± s.d. from three independent experiments.

## 2.11. Statistical analysis

The data are presented as the mean ± s.d. from at least three independent experiments, with the experiments being performed in at least triplicate. Statistical differences were analysed by one-way ANOVA, followed by the *post hoc* Bonferroni test, with $p < 0.05$ considered statistically significant. The one-way ANOVA and Bonferroni test in the SPSS software were used for the analyses.

# 3. Results

## 3.1. GGOH significantly reversed ZA-suppressed MSC viability and proliferation

The effect of GGOH on MSC viability was first determined and the results showed that GGOH at concentrations less than 100 µM (following 3 days in culture) and 75 µM (following 7 days in culture) appeared to have no discernible effects on MSC viability and proliferation, whereas the higher doses tested significantly retarded the proliferation of MSCs (figure 1a,b, respectively). The concentration of ZA used in this study was based on our previous report [8] and on the level detected in bone from patients with existing osteonecrosis (0.4–4.6 µM) [6]. The results in figure 1c show that following 3 days in culture, a discernible inhibitory effect of 5 µM ZA on the number of viable MSCs was evident, and far fewer viable cells were found when 100 µM GGOH was also added in the culture. This confirmed the cytotoxicity of GGOH at a high dose. GGOH at concentrations between 5 µM and 25 µM seemed to reverse the cytotoxicity of ZA. The cytoprotective role of GGOH was more pronounced in a longer culture time (7 days), which demonstrated that treatment of MSCs with GGOH at 10–25 µM preserved high numbers of viable cells exposed to ZA. By contrast, dramatic cell death was observed when MSCs were exposed to ZA alone or together with GGOH at low doses of 0.1–1 µM and a high dose at 100 µM (figure 1c). The results suggested that only an optimal range of GGOH dose had a cytoprotective effect against ZA cytotoxicity. The results in figure 1d show that after ZA treatment for 3 days, the percentage of viable cells decreased to 64% of that in the untreated control cells, and GGOH at concentrations between 10 and 15 µM significantly reversed the suppression effect on ZA on the MSC viability and proliferation. The protective effect of GGOH was clearly demonstrated at Day 7 in culture when GGOH (5–75 µM) significantly recued MSC viability and proliferation, with the concentrations between 10 and 15 µM being most effective (figure 1e). With these concentrations of GGOH, the cells viability was markedly restored from 20% to 80% of that of the control cells. A progressing decrease in cell viability, from Day 3 to Day 7, of cells treated with ZA and 100 µM GGOH confirmed the lack of cytoprotective role of GGOH at high concentrations observed on Day 3. The effect of GGOH at this optimal concentration, i.e. 10 µM, on ZA cytotoxicity reversal under a higher dose of ZA of 50 µM was also evident, as shown in figure 1f. It is noteworthy that GGOH alone had little, if any, effect on the viability and proliferation of MSCs.

The concentration of GGOH at 10 µM appeared to be optimal, protecting MSCs from the cytotoxic effect of a range of 5–50 µM of ZA, and was thus used in the subsequent experiments to investigate the mechanisms involved in its cytoprotective role against ZA.

## 3.2. GGOH-protected MSCs underwent osteogenic differentiation and formed biomineralization

GGOH-rescued ZA-treated MSCs formed biomineralization comparable to that produced by normal MSCs at both 14 and 21 days under osteogenic induction (figure 2a). The expression of osteogenic genes related to osteoblast differentiation, i.e. RUNX2, COL-I, ALP and OCN, was also examined, and the results in figure 2b show that under osteogenic induction, all the genes were expressed two–threefold higher than those in control cells. In addition, GGOH-protected MSCs, that were rescued from a high-dose ZA treatment, under osteogenic induction, expressed comparable levels of these genes to the cells previously not exposed to ZA (figure 2b). GGOH-treated MSCs (without ZA treatment) underwent osteoblast differentiation and produced biomineralization comparable to those of normal MSCs (figure 2a); ZA pre-treatment resulted in extremely reduced viable MSC cells which were not recovered when further cultured in OM (data not shown). Therefore, assays were not possible for this group. Similar gene expression responses were obtained from all three biological replicates. The results indicated that GGOH-treated MSCs that were rescued from the toxicity of ZA could differentiate into osteoblasts and formed biomineralization, suggesting a possible clinical benefit of the cytoprotective role of GGOH against ZA toxicity.

## 3.3. GGOH reversed ZA-mediated Ki67$^+$ MSC number reduction, cell cycle arrest and apoptosis

It is possible that GGOH rescued ZA-suppressed MSC viability and proliferation via regulation of cell cycle and apoptosis. To investigate mechanism(s) of the reversal effect of GGOH on ZA-induced cell death, changes in the number of proliferating cells, cell cycle and apoptosis were investigated. The

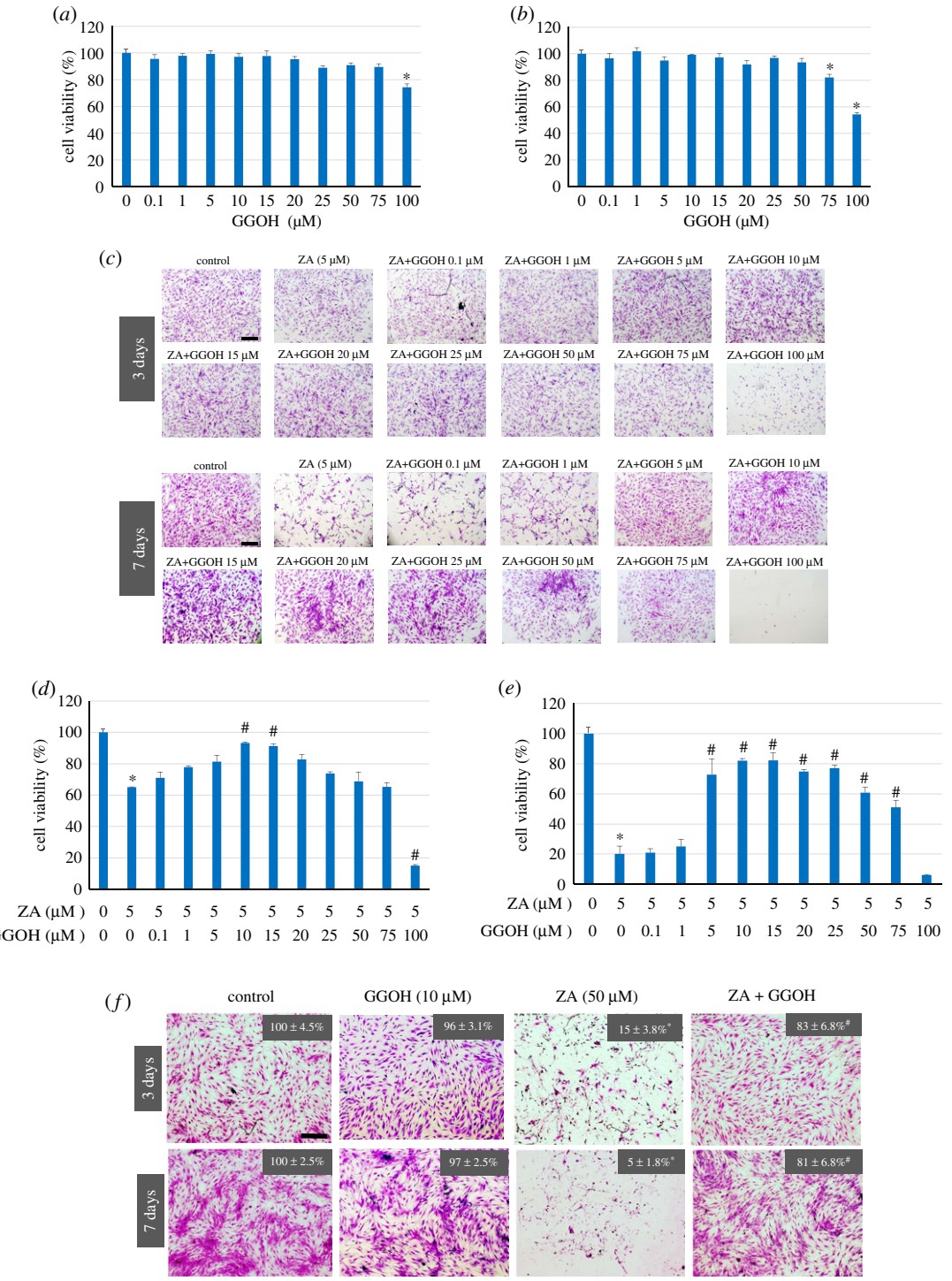

**Figure 1.** Effects of GGOH on MSC viability and ZA-suppressed MSC viability. MSCs were cultured with GGOH (0–100 μM) for 3 (*a*) and 7 (*b*) days and the MSC viability was measured by MTT, as described in the Material and methods. In (*c*–*f*), MSCs were cultured with ZA (0, 5 or 50 μM) and GGOH (0–100 μM) for 3 and 7 days. The cells were stained with crystal violet (*c*), and the MSC viability was measured by MTT (*d*,*e*), as described in the Materials and methods. An optimal concentration of GGOH (10 μM) was selected to examine its ability to prevent MSC death induced by a high dose of ZA (50 μM) for 3 and 7 days using crystal violet staining (*f*). Cell viability is expressed as the mean percentage of control (100%). The results shown in (*a*,*b*) and (*d*–*f*) are presented as the mean percentage ± s.d. from three independent experiments. The values in (*f*) are derived from MTT assay. *$p < 0.05$ versus control cells without treatment; #$p < 0.05$ versus cells with ZA treatment. Scale bar = 10 μm.

results confirmed the cytoprotective effect of GGOH on ZA cytotoxicity following 1 day in culture, as shown in figure 3*a*. Flow cytometry (FCM) analysis of a cell proliferation marker Ki67 demonstrated that ZA decreased Ki67$^+$ proliferating cells from 83% in the control cells to 63% in ZA-treated cells;

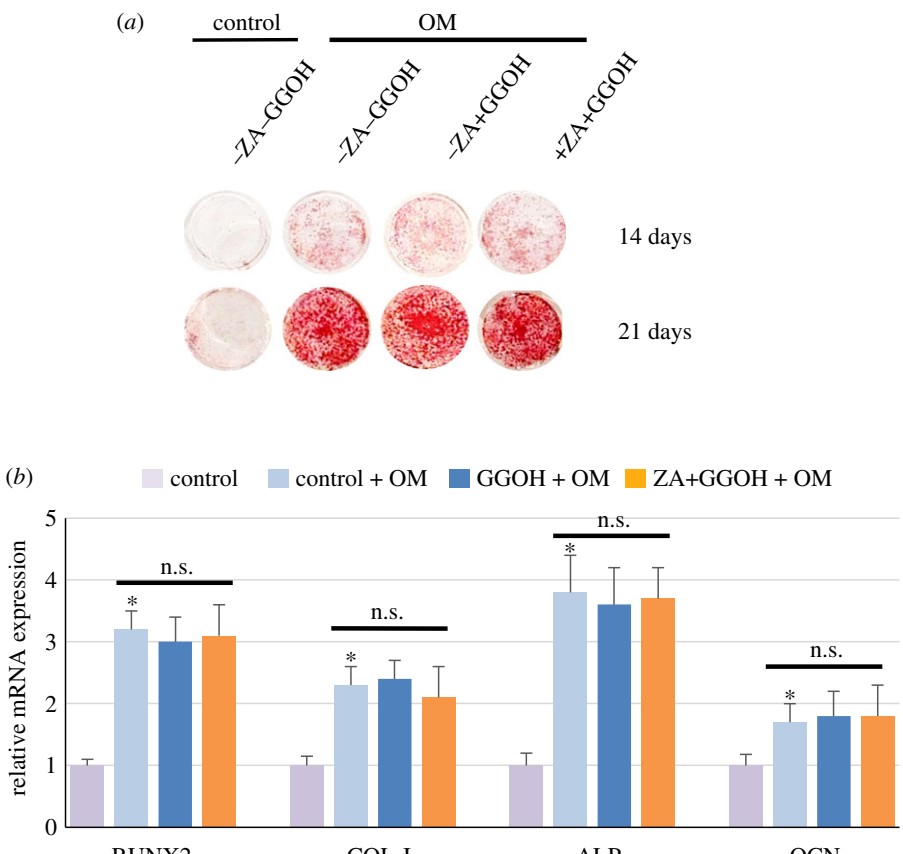

**Figure 2.** Osteoblast differentiation of GGOH-protected MSCs. MSCs were cultured with ZA (50 µM) and GGOH (10 µM) as indicated for 7 days, and GGOH-protected MSCs were subsequently further induced in OM for 14–21 days in the absence of ZA and GGOH. The formation of mineralization and the expression of osteogenic genes were examined by alizarin red S staining on Days 14 and 21 (*a*) and by Q-PCR on Day 14 (*b*), respectively. The biomineralization data shown are representative of three independent experiments. The mRNA expression is expressed as mean fold-change of control (1.0), and the data are presented as the mean fold-change ± s.d. from three independent experiments. $^{*}p < 0.05$ versus control cells without treatment; n.s.: not significant.

however, GGOH raised the proportion of Ki67$^{+}$ cells (80%) to the level near the control cells (figure 3*b*). Among Ki67$^{+}$ dividing cells, representative histograms in figure 3*c* show a decrease in cells in S phase and a larger proportion of cells in G2/M phase in the ZA group, compared with those in the control cells, and GGOH appeared to reverse such effects from ZA treatment. A summary of cell cycle distribution shown in figure 3*d* confirmed that while the proportions of cells in G1 phase were not different among the groups, ZA treatment significantly decreased the proportion of cells in S phase by 31%, and GGOH reversed this by increasing cells in S phase to the level of 92% of the control cells. In contrast with the effect on S phase distribution, approximately 13% of divining MSCs treated with ZA were in G2/M phase, whereas only 7% and 9% of control cells and of cells treated with ZA and GGOH, respectively, were in G2/M phase (figure 3*d*). In addition, a sevenfold increase in apoptosis was observed in ZA-treated cells compared with that in the control MSCs (12.2% versus 1.7%), and GGOP significantly inhibited ZA-induced MSC apoptosis, resulting in only 3.4% apoptosis (figure 3*e*). Taken together, although GGOH alone had very little, if any, effect on Ki67$^{+}$ MSC number, cell cycle arrest and apoptosis, it significantly reversed ZA-mediated Ki67$^{+}$ MSC number reduction, cell cycle arrest and apoptosis.

## 3.4. GGOH prevented ZA-inhibited YAP activation

Since YAP activation, which is crucial for cell viability and proliferation, can be regulated by protein geranylgeranylation [25], it is possible that the alteration of geranylgeranylation by ZA and GGOH may control the activation of YAP. Therefore, the effects of ZA and GGOH on YAP nuclear localization and the phosphorylation level at serine 127 of YAP were examined. Representative

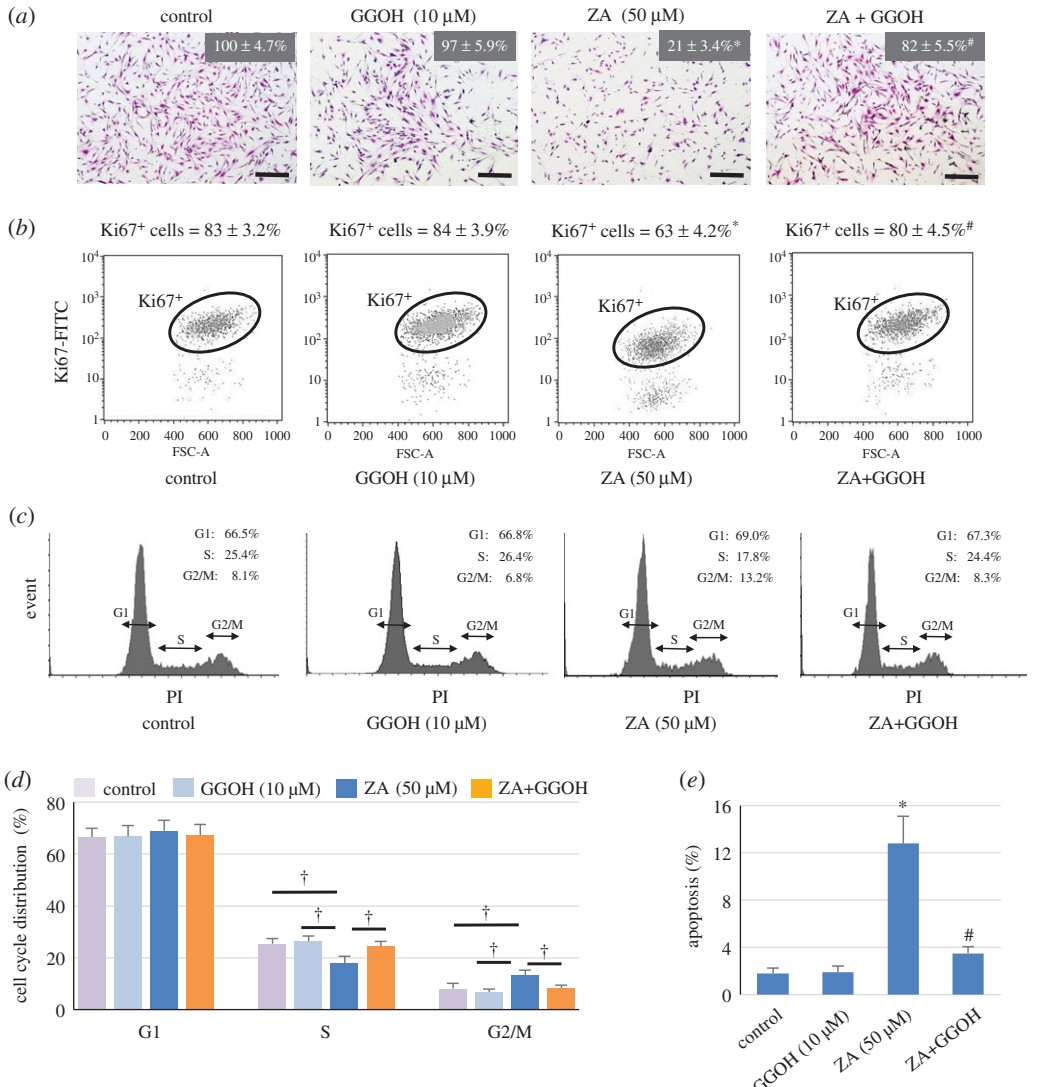

**Figure 3.** Effects of GGOH on ZA-suppressed cell viability and cell cycle. MSCs were treated with ZA (50 μM) and GGOH (10 μM) as indicated for 24 h. Cells were stained with crystal violet to visualize cell morphology (*a*), and the cell viability determined by MTT is also shown as the mean percentage of control (100%). Ki67$^+$ cells, cell cycle analysis and apoptotic cells were determined using FCM, as described in the Material and methods, and the results are shown in (*b*), (*c*,*d*)) and (*e*), respectively. Noted that among Ki67$^+$ cells, the proportions of each phase in cell cycles were determined, and the representative cell cycle histograms are shown in (*c*), and a summary of the results is shown in (*d*). The results in (*a*,*b*) and (*d*,*e*) are expressed as the mean percentage ± s.d. from three independent experiments. $^*p < 0.05$ versus control cells; $^\#p < 0.05$ versus cells treated with ZA only. $^\dagger p < 0.05$. Scale bar = 10 μm.

immunofluorescence staining of YAP in figure 4*a* shows dominant nuclear localization of YAP in untreated control cells and GGOH-treated cells. In marked contrast, intracytoplasmic distribution of YAP was mainly observed in cells treated with ZA. However, when ZA-treated cells were supplemented with GGOH, YAP staining appeared to be localized both in nucleus and cytoplasm (figure 4*a*). Quantitatively, approximately 70% and 75% of control cells and GGOH-treated cells, respectively, showed pronounced YAP nuclear localization. Only 30% of cells showed predominant nuclear YAP staining in the presence of ZA (figure 4*b*). The addition of GGOH to ZA-treated cells increased the level of cells with nuclear YAP to approximately 60% of the total cells. The results in figure 4*c*,*d* show that ZA markedly increased phosphorylated YAP by approximately 2.5-fold. Although GGOH had little effect on the level of phosphorylated YAP, it significantly reduced ZA-induced YAP phosphorylation. However, the expression of YAP was not significantly modulated by either GGOH or ZA. The results suggested that GGOH was able to prevent ZA-inhibited YAP activation in MSCs by regulating nuclear translocation and phosphorylation of YAP.

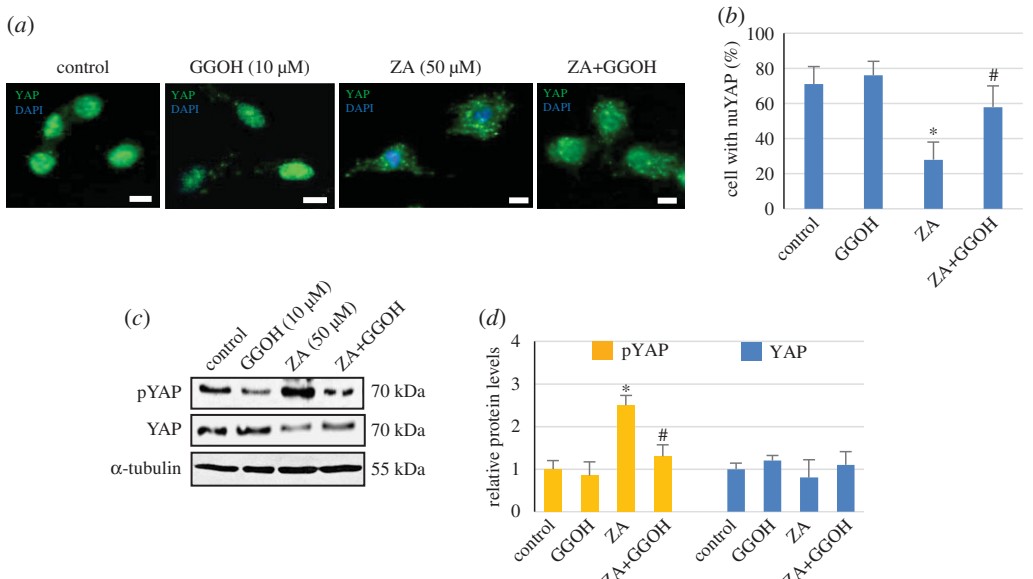

**Figure 4.** Effect of GGOH on ZA-suppressed YAP activation. MSCs were treated with ZA (50 µM) and GGOH (10 µM) as indicated for 24 h, and the confocal immunofluorescence of YAP was analysed, as described in the Material and methods. Representative immunofluorescence staining of YAP (green) and DAPI (blue) in untreated, GGOH-treated, ZA-treated and ZA and GGOH-treated cells are shown in (*a*), and a summary of proportions of YAP nuclear translocation is shown in (*b*). The results are expressed as the mean percentage of cells with nuclear YAP (nuYAP) staining ± s.d. from three independent experiments. In (*c*), the expression of phosphorylated YAP (pYAP) and YAP proteins in cells treated with ZA (50 µM) and GGOH (10 µM) as indicated for 24 h was determined by Western blot analysis, and the representative immunoblots from three independent experiments are shown in (*c*). Levels of α-tubulin are shown for comparison as loading controls. The relative protein levels of pYAP and YAP were normalized to that of α-tubulin, and the data are presented as the mean fold-change of the untreated control (1.0) ± s.d. from three independent experiments (*d*). $^{*}p < 0.05$ versus control cells; $^{#}p < 0.05$ versus cells treated with ZA only. Scale bar = 5 µm.

## 3.5. Protective effect of GGOH on ZA-suppressed MSC viability involved Rho activity and activation of YAP

ZA is known to inhibit geranylgeranylation of many proteins, including Rho family GTPases, and GGOH increases geranylgeranylated protein levels, and thus stimulates the activity of Rho GTPases. The Rho subfamily itself (RhoA, B and C) plays an important role in regulating stress fibres and in turn controlling nuclear translocation of various signalling proteins, including that of YAP [28,29]. We confirmed the involvement of Rho subfamily proteins in the cytoprotective effect of GGOH by first measuring the activity of the active form of RhoA, i.e. GTP-bound RhoA, in ZA-treated samples, and the results showed that the RhoA-GTP level in cells with ZA was only 24% of that in the untreated control cells (figure 5*a*). Although GGOH did not alter the level of RhoA-GTP, it significantly restored the level of RhoA-GTP to approximately 79% of the control cells in the presence of ZA, confirming the stimulatory effect of GGOH on the RhoA activity in ZA-treated cells (figure 5*a*). Moreover, when the activity of Rho subfamily proteins was blocked by a chemical inhibitor Rhosin, GGOH-restored YAP nuclear translocation was partly inhibited by approximately 70% (figure 5*b*). A chemical inhibitor and an activator of YAP (DH and LPA, respectively) were first used to determine its role in mediating the cytoprotective effect of GGOH, while these chemicals suppressed and stimulated the cell viability to the levels of 49% and 150% of that of the control, respectively. DH partly, but significantly, blocked the cytoprotective ability of GGOH against ZA (figure 5*c*). Increased YAP activation by LPA also restored the viability of cells treated with ZA, but to a lesser extent compared with GGOH treatment (59% of the control in the ZA+LPA group versus 82% of the control in the ZA+GGOH group), suggesting that YAP might be involved in the protective effect of GGOH. To unequivocally establish the role of YAP, siRNA specific to YAP was used, and YAP protein expression was suppressed by more than 50% compared with that expressed in cells transfected with control siRNA (figure 5*d* and *e*). In YAP siRNA-transfected cells, GGOH was able to reverse ZA-suppressed MSC viability, but its reversal effect was significantly lower than that observed in the control siRNA-

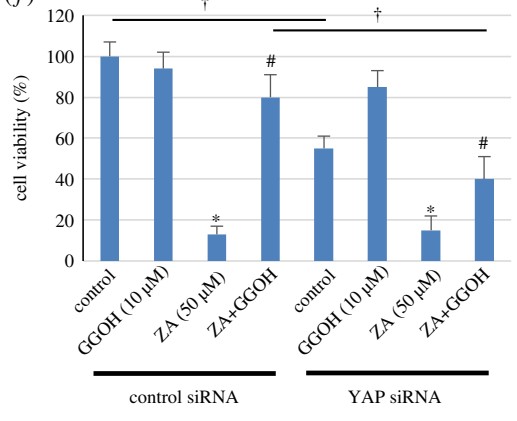

**Figure 5.** Involvement of Rho and YAP in the protective effect of GGOH against ZA cytotoxicity. In (*a*), MSCs were treated with ZA (50 µM) and GGOH (10 µM) as indicated for 24 h, and RhoA activity was assayed, as described in the Material and methods. The RhoA activity is expressed as the mean percentage of control (100%) ± s.d. from three independent experiments. In (*b*), cells were treated with 30 µM Rhosin prior to the addition of ZA (50 µM) and GGOH (10 µM) as indicated, and after 24 h, cells were immunostained for YAP, and the proportion of cells with nuYAP immunofluorescence staining was analysed, as described in the Material and methods. The results are expressed as the mean percentage of cells with nuclear YAP (nuYAP) staining ± s.d. from three independent experiments. In (*c*), cells were pretreated with 20 µM DH or 10 µM LPA in addition to ZA (50 µM) and GGOH (10 µM) as indicated, and after 3 days, MTT assay was used to determine the number of viable cells. The results are expressed as the mean percentage of untreated control cells (100%) ± s.d. from three independent experiments. Silencing of YAP protein was confirmed by Western blot analysis after 48 h of siRNA transfection, and a representative immunoblot (from three independent experiments) of YAP expression is shown in (*d*). Levels of α-tubulin are shown for comparison as loading controls. In (*e*), the protein level of YAP normalized to that of α-tubulin is expressed as the mean fold-change of non-transfected control (1.0) ± s.d. from three independent experiments. In (*f*), cells were transfected with control or YAP siRNAs, and 48 h post-transfection, cells were further cultured with ZA (50 µM) and GGOH (10 µM) as indicated for 3 days. The MSC viability was measured by MTT, and the data are expressed as mean percentage of non-transfected cells (100%) ± s.d. from three independent experiments. $*p < 0.05$ versus (siRNA-transfected) control cells (within the same siRNA group); $^{#}p < 0.05$ versus (siRNA-transfected) cells treated with ZA only (within the same siRNA group); $^{$}p < 0.05$ versus ZA+GGOH group; $^{†}p < 0.05$.

transfected cells, suggesting that YAP is involved in protective effect of GGOH. The results also showed that the viability of YAP siRNA-transfected MSCs was significantly lower than that of the control siRNA-transfected cells, suggesting that YAP is essential for MSC viability in normal culture without ZA treatment.

## 3.6. Protective effect of GGOH on ZA-suppressed MSC viability was mediated through CDK6

Cell cycle arrest at G1/S phase transition is regulated by a number of cell cycle check-point proteins [30,31], and CDK4/6, but not CDK2, activity, which is required throughout G1 phase, is sufficient for cells to enter S phase [32], suggesting an important role of CDK4/6 activity in cell viability and proliferation. We, therefore, determined the involvement of CDK4/6 in the protective effect of GGOH on ZA-suppressed MSC viability by using the CDK4/6 inhibitor palbociclib. In figure 6a, the results confirmed that the MSC viability was significantly reduced by ZA and palbociclib to 18% and 38% of the control group, respectively. Palbociclib significantly inhibited the reversal effect of GGOH on ZA-suppressed MSC viability, resulting in cell viability of approximately 40% of the control group, which remained higher than that of cells treated with ZA alone (figure 6a). The results in figure 6b show that while the expression of CDK4 protein was not influenced by any of the treatments, the expression of CDK6 was strongly reduced by ZA, which was rescued by GGOH. Quantitative analysis of the results demonstrated that none of the treatments altered CDK4 protein expression (figure 6c), but ZA suppressed CDK6 protein expression to the level of approximately 55% of that expressed in the control cells (figure 6d). This inhibitory level was reversed by GGOH, resulting in CDK6 expression of about 80% of that expressed in the control cells (figure 6d). In addition, the expression of CDK6 appeared to be transcriptionally regulated by ZA which downregulated the expression of CDK6 mRNA, and GGOH also rescued this at the transcriptional level (figure 6e). The results also showed that the expression of CDK6 mRNA in YAP siRNA-transfected MSCs was significantly lower than that in the control siRNA-transfected cells (figure 6f), suggesting that the basal expression level of CDK6 mRNA was regulated by YAP. Knocking down of YAP also decreased the reversal effect of GGOH on ZA-suppressed CDK6 mRNA expression (figure 6f). In YAP siRNA-transfected cells, GGOH was able to reverse ZA-suppressed CDK6 mRNA, but its reversal effect was significantly lower than that observed in the control siRNA-transfected cells. The results suggested that GGOH-rescued ZA-suppressed CDK6 expression was mediated in part by YAP. Taken together, the results indicated that GGOH-rescued ZA-suppressed MSC viability was mediated at least partly by CDK6.

## 4. Discussion

When prescribed ZA for more than 3 years, patients may have an accumulated level of ZA in their bone which can be very high and highly cytotoxic [33], possibly diminishing the pool of MSCs that help regenerate patients' injured bone. During the healing of bone injury, free unbound ZA is expected to be high at the microenvironment where the injured bone takes place. A high concentration of ZA reduces the number of viable MSCs and suppresses the regeneration of new bone. Previous studies suggested the role of viable MSCs in preventing and treating defective bone healing induced by ZA [34,35], emphasizing defective osteogenesis as an important factor in developing MRONJ. The present *in vitro* study has shown, for the first time, that in MSCs, the reversal effect of GGOH on ZA cytotoxicity was at least partly mediated through Rho-dependent activation of YAP. A summarized mechanism of action of GGOH is depicted in figure 7. We propose that in patients receiving long-term ZA, local delivery of GGOH may help preserve a pool of viable MSCs that retain their osteogenic potency, thus helping prevent or treat MRONJ.

Notably, N-BP-related defective bone healing in animal models was successfully treated by systemic infusion of MSCs [34,35]. Nevertheless, in cancer patients, systemic infusion of MSCs may promote cancer metastasis and recurrence [36,37]. In addition, disseminated intravascular thrombosis and prolonged prothrombin time have been reported for systemic MSC infusion which involves anticoagulant therapy [38]. It is proposed that GGOH can be locally and directly administrated using a biocompatible carrier into the necrotic bone defect, or the extraction socket, of the jaw bone to treat, or prevent, MRONJ. The local application of GGOH therapy is therefore beneficial especially in cancer patients as this method helps circumvent the aforementioned side effects (from systemic delivery of MSCs) as well as avoids GGOH interference of osteoclast-inhibiting clinical benefit of N-BPs that is required in other sites of skeletal bones. The presence of GGOH within a localized area in the jaw

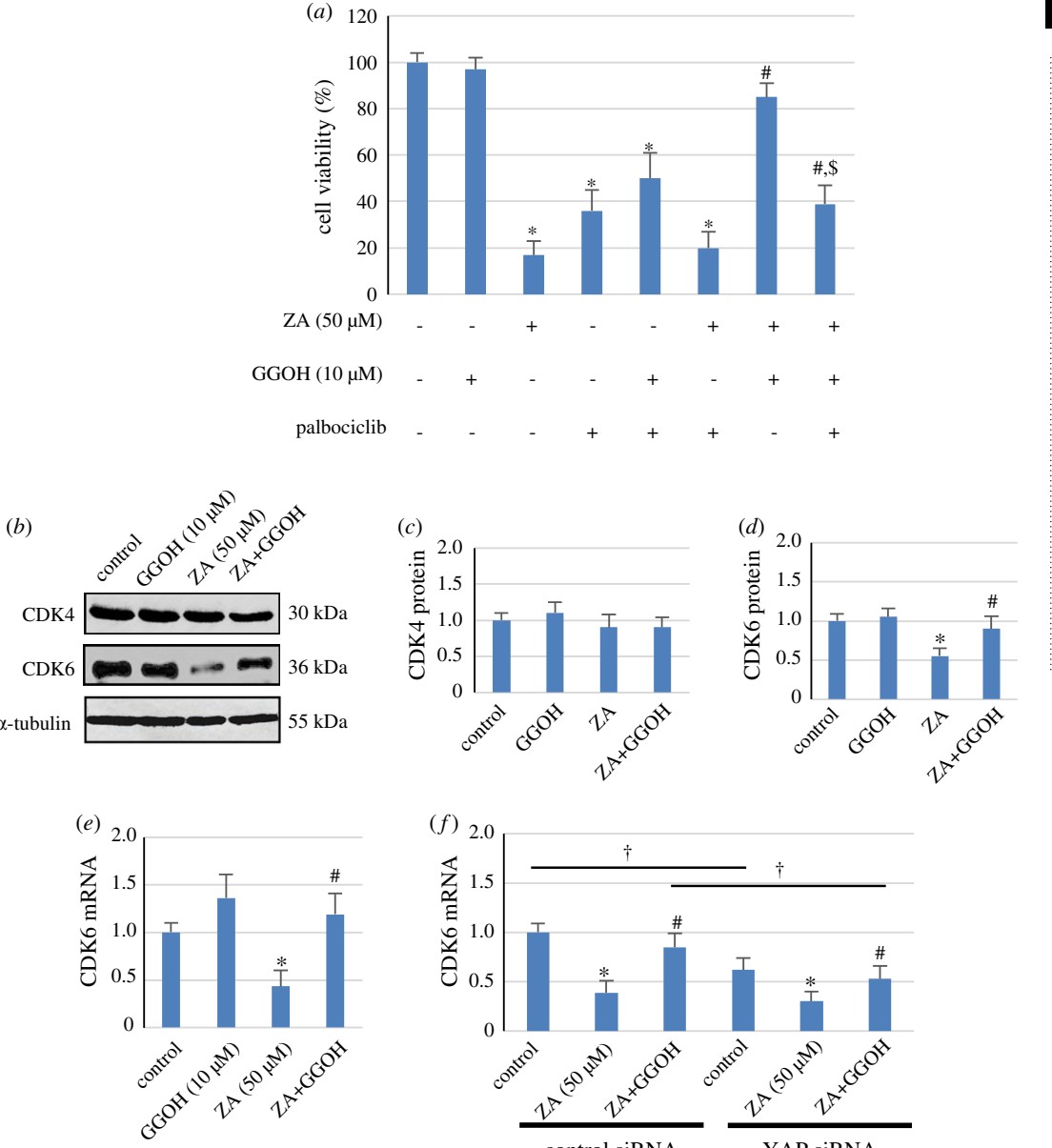

**Figure 6.** Involvement of CDK4/6 in the protective effect of GGOH against ZA cytotoxicity. In (*a*), MSCs were pretreated with 1 μM palbociclib for 30 min prior to the addition of ZA (50 μM) and GGOH (10 μM) as indicated, and after 3 days, MTT assay was used to determine the number of viable cells. The results are expressed as the mean percentage of untreated control cells (100%) ± s.d. from three independent experiments. The expression of CDK4 and CDK6 proteins in cells treated with ZA (50 μM) and GGOH (10 μM) as indicated for 3 days was determined by Western blot analysis. The representative immunoblots shown in (*b*) are from 3 independent experiments. The relative protein levels of CDK4 and CDK6 were normalized to that of α-tubulin, and the data presented as the mean fold-change of the untreated control (1.0) ± s.d. from three independent experiments are shown in (*b*) and (*c*), respectively. The expression of CDK6 mRNA in cells treated with ZA (50 μM) and GGOH (10 μM) as indicated for 24 h was determined by Q-PCR (*e*). In (*f*), cells were transfected with control or YAP siRNAs for 48 h, followed by ZA (50 μM) and GGOH (10 μM) as indicated for 24 h, and the mRNA expression of CDK6 was examined by Q-PCR. The mRNA expression results are expressed as the mean fold-change of control (1.0) ± s.d. from three independent experiments. $^*p < 0.05$ versus (siRNA-transfected) control cells (within the same siRNA group); $^\#p < 0.05$ versus (siRNA-transfected) cells treated with ZA only (within the same siRNA group); $^\$p < 0.05$ versus ZA+GGOH group; $^\dagger p < 0.05$.

bone will help preserve the viability of MSCs against the released unbound form of ZA during the bone repair without interfering in the effect of ZA in other skeletal bones such as femur and hip. Diffusion of the locally administrated GGOH to these distant bones is possible, but its concentration may not be high enough to elicit any effects against the entire ZA in the body. GGOH, an alcohol derivative of an endogenously produced intermediate GGPP, is a mevalonate pathway activating molecule that is

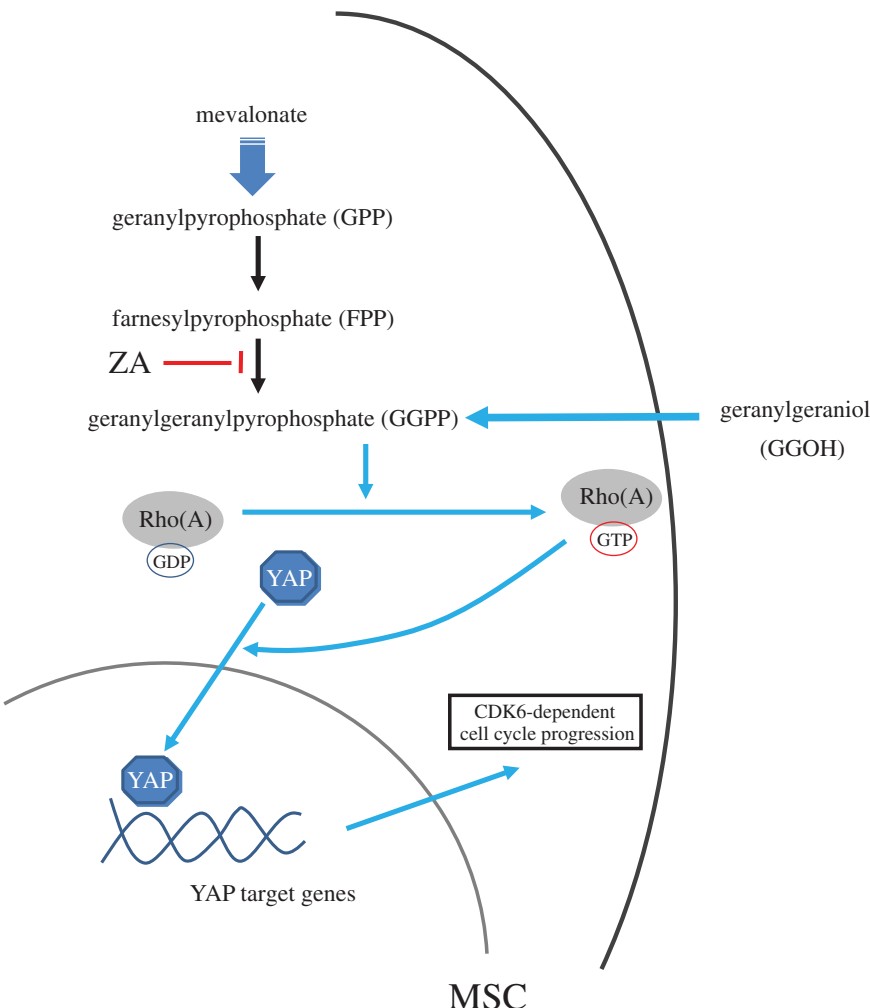

**Figure 7.** A proposed mechanism of the cytoprotective effect of GGOH against ZA cytotoxicity for prevention and treatment of MRONJ. The cytoprotective effect of GGOH prevents ZA cytotoxicity, thus maintaining a pool of viable MSCs with osteogenic potency, by rescuing the function of decreased GGPP. In MSCs exposed to ZA, GGOH increases RhoA activity and activates YAP nuclear translocation. GGOH-induced YAP activation increases CDK6-dependent cell viability and cell division. We propose that considering the reversal effect of GGOH against ZA cytotoxicity, local delivery of GGOH could help preserve a pool of viable MSCs that retain their osteogenic potency, thus helping prevent or treat MRONJ in patients receiving long-term ZA administration.

intracellularly converted to GGPP in humans [39] with a crucial role in controlling protein prenylation [15,16]. The present mechanism of action study provided the first evidence supporting that GGOH-prevented MSC death from ZA cytotoxicity was at least partly mediated through Rho-dependent activation of YAP. Rho subfamily proteins and YAP have been shown to play important roles in viability, proliferation and stemness of MSCs [20,21,23,24].

The present results demonstrated the involvement of Rho subfamily proteins in the cytoprotection of GGOH against ZA and suggested that among the three isoforms in the Rho subfamily, RhoA could play a role as its reduced activity caused by ZA was reversed by GGOH. However, we cannot rule out the possibility that RhoB and RhoC may also be involved in the reversal effect of GGOH. Rhosin, a small-molecule compound used in the present study, inhibits the activity of RhoA, RhoB and RhoC. The use of a second pharmacological inhibitor of the RhoA pathway or interference with RhoA activity in another manner (e.g. targeting the expression of guanine nucleotide exchange factors (GEFs), GTPase-activating proteins (GAPs), guanine-dissociation inhibitors and active/inactive RhoA) will undoubtedly establish the functional importance of RhoA specifically in GGOH-mediated cytoprotection against ZA in MSCs. It is also noteworthy that some effects of ZA may also be Rho-independent as Rho proteins are not the only geranylgeranylated proteins in MSCs. Ras, Cdc42 and Rac GTPases and other types of proteins, such as heterotrimeric G proteins, are also

geranylgeranylated. Decreased prenylation of other small GTPases, such as Rap, Ras and Cdc42, has been linked to some effects of ZA [40–43]. These proteins are essential for multiple cellular processes, including cell proliferation, cell movement, cytoskeletal rearrangement and apoptosis [19,44,45]. Whether or not these molecules are involved in the protective effect of GGOH on ZA-induced MRONJ requires further studies. In addition, the identification of the proteins that became unprenylated after ZA treatment should be of great interest in understanding the interaction of signalling pathways triggered by ZA and rescued by GGOH.

The concentrations of ZA used in this study were based on our previous report [8] and on the levels detected in bone *in vivo* (ranging between 1 and 100 µM) [4–6]. Increasingly accumulated ZA in the bone of patients with long-term use of ZA is expected, and the concentrations can be very high, possibly up to 100 µM [4,5]. Our previous study showed that the concentrations of ZA ranging between 50 µM and 100 µM appeared to be highly cytotoxic dose at a concentration of 100 µM [8] and the concentration at 50 µM was thus selected in this study. The present study showed that cytotoxicity of ZA at low and high concentrations (5 and 50 µM, respectively) was effectively prevented by GGOH at approximately 10–15 µM, which provided maximum protection from cytotoxicity of ZA at low and high concentrations (5 and 50 µM, respectively). These concentrations of GGOH (10–15 µM) were much lower than its cytotoxic dose (100 µM), suggesting that GGOH could be used safely and effectively in humans. Further *in vivo* toxicity studies of GGOH are essentially required.

In the present study, GGOH was unable to completely prevent ZA cytotoxicity, with approximately 80% of MSCs being rescued. A possible explanation could be that in addition to its suppression on GGPP production, ZA inhibits a mevalonate pathway enzyme downstream to an intermediate isopentenyl pyrophosphate, which can be converted to a cytotoxic mediator triphosphoric acid 1-adenosin-5′-yl ester 3-(3-methylbut-3-enyl)ester (ApppI) [46,47]. The production of ApppI is, therefore, unaffected by GGOH. However, the rescued MSCs retained their osteogenic function by forming bone-like mineralization, when induced, comparable to cells naive to ZA, suggesting a clinical importance of these GGOH-protected MSCs. It is also important to note that ApppI inhibits the mitochondrial adenine nucleotide translocase (ANT), thus resulting in cell apoptosis [48]. The expression of ANT can be regulated by various intracellular signalling pathways initiated by growth factors and cytokines [49,50]. It is therefore possible that during an initial phase of bone healing, a wide range of anabolic mediators involved in the proliferation and survival of stem/progenitor cells may modulate the mitochondrial energy metabolism and expression of ANT, and thus protecting MSCs from ApppI-induced cytotoxicity. Further studies are required to prove this hypothesis and identify key molecules that effectively prevent ApppI-mediated cytotoxicity induced by ZA.

YAP is the major downstream effector of the Hippo pathway [51], and when activated, YAP molecules translocate into the nucleus and activate its target genes, including those involved in cell proliferation and cell cycle progression [23,24]. By contrast, when phosphorylated at Ser127, phosphorylated YAP accumulates in the cytoplasm, ultimately leading to YAP degradation [20]. It has been shown that geranylgeranylation downstream of the mevalonate pathway is required and is sufficient to activate YAP and activate ERK, which thus stimulates cell proliferation [52]. The present results demonstrated that GGOH-mediated YAP activation reversed ZA-induced cell cycle arrest by stimulating cell cycle progression and promoting more cells to enter S phase and G2/M phase, as previously reported for the role of YAP in the induction of cell cycle progression in chondrocytes [53]. Moreover, YAP has been shown to stimulate proliferation and inhibit apoptosis of stem cells [21,22]. However, it is also possible that in this context, other downstream effectors of Rho can also mediate cell division and proliferation of MSCs. These potential effectors include stress fibre actin, cyclin D1 and cyclin-dependent kinase inhibitors p21 and p27 [20,54]. Moreover, Rho-independent regulators of YAP, such as PDZ-binding kinase, IGF-1 and Wnt signallings [54–56], could also contribute to the cytoprotection of GGOH. Further studies are needed to advance our knowledge about the mechanisms controlling GGOH effect against ZA cytotoxicity.

Regulation of cell cycle entry is critical for the growth, repair and maintenance of mammalian tissues. Mitogens can induce cells to enter the cell cycle by exiting quiescence (G0 phase) to enter G1 phase before replicating their DNA in S phase and undergoing cell division in mitosis. Cell cycle progression is regulated by the activity of CDKs, such as CDK4, CDK6 and CDK2, and several important mechanisms are involved in the regulation of CDKs and their partner cyclins' activity. It has been shown that CDK4/6, but not CDK2, activity, which is required throughout G1 phase, is sufficient for cells to enter S phase [32]. In the present study, by using the CDK4/6-specific inhibitor palbociclib [32,57], the results suggested that GGOH cytoprotection against ZA-suppressed MSC viability and proliferation was, at least in part, CDK4/6-dependent. However, ZA decreased the

expression of CDK6, but not CDK4, at protein and mRNA levels, and this inhibitory effect was rescued by GGOH. This is consistent with the presence of known YAP/TEAD binding elements in the promoter of CDK6, but not CDK4, gene, and CDK6 is known to be a direct downstream target of YAP in the regulation of cellular senescence [58]. Using RNA interference specific to YAP, the present results unequivocally establish that YAP-mediated upregulation of CDK6 contributed to the cytoprotective role of GGOH against ZA in MSCs. This is important for efficient S phase entry which is essential for tissue repair and healing. In addition to Rho GTPase-YAP signalling shown in the present study, other important signalling pathways, such as those initiated by mitogens, via MAPK/cyclin D1, and by DNA damage, via p53 and p21, are known to regulate CDK4/6 activity [59]. The effects of ZA and GGOH on the upstream regulators and potential effectors of CDK6 in MSCs remain unknown and warrant further studies.

Due to the ubiquitous expression of a majority of Rho GTPases and their implication in fundamental cellular processes, their endogenous expressions and activities must be tightly regulated to ensure that cell division progresses properly without tumour transformation. Dysregulation of Rho GTPase signalling is closely associated with tumorigenesis and malignant phenotypes [60]. It has been shown that the protein isoprenylation-dependent mechanism governing Rho GTPases turnover may represent a mechanism by which cells maintain a proper level of Rho-dependent cell signalling [61–63]. Moreover, some less understood post-translational modifications, such as sumoylation, ubiquitylation, AMPylation and transglutamation, may also regulate the activity of Rho GTPases [64]. In the present study, exogenously supplemented GGOH alone did not increase the level of endogenous Rho activity, YAP activation, CDK6 and the number of viable MSCs. However, when the activity of Rho was suppressed by ZA, GGOH stimulation of Rho activity, YAP activation, CDK6 and the number of viable MSCs became evident. The precise mechanism(s) underlying this remain(s) to be investigated. It is possible that negative feedback control could occur in response to the increased basal level of endogenous geranylgeranylation by the GGOH treatment alone. When the level of endogenous geranylgeranylation is reduced by ZA, exogenously added GGOH could help maintain the basal level of geranylgeranylation required for fundamental cellular processes, such as cell cycle progression. Moreover, a narrow range of GGOH concentrations that are optimal for cytoprotection against ZA observed in the present study might be attributed to a number of highly regulated mechanisms for controlling Rho GTPase-mediated cellular functions in MSCs. This could be responsible for the prevention of uncontrolled cell cycle progression and cell proliferation by the exogenously added GGOH. Careful *in vivo* determination of an effective dose level is required to obtain its accurate dose–response relationship. The *in vitro* information will help support its safety for future animal and clinical studies.

# 5. Conclusion

The present study has shown, for the first time, that in MSCs, the cytoprotective effect of GGOH against ZA cytotoxicity was at least partly mediated through Rho-dependent activation of YAP, which, in turn, promoted CDK6-mediated cell viability and proliferation. The present findings support GGOH as an attractive pharmacological compound for maintaining MSC viability against ZA and identify YAP as a potential therapeutic target for ZA-induced MRONJ. However, further studies, especially *in vivo* studies, are required to test the effectiveness of the administration of GGOH with biocompatible delivery methods.

Data accessibility. Data are available from the Dryad Digital Repository at https://doi.org/10.5061/dryad.wstqjq2jq [65].
Authors' contribution. W.S. contributed to conception and design, acquisition, analysis and interpretation of data, and drafted and critically revised the article for important intellectual content. W.H. contributed to the acquisition, analysis and interpretation of data. W.J. contributed to the conception and design, interpretation of data and critically revised the article for important intellectual content. All authors approved the final version of the article and agreed to be accountable for all aspects of the work in ensuring that questions related to the accuracy or integrity of any part of the work are appropriately investigated and resolved.
Competing interests. We declare we have no competing interests.
Funding. The present study was funded by Thammasat University Research Fund (Contract no. TUGR 2/54/2562) to W.S., National Metal and Materials Technology Center (MTEC) (grant no. P1650068 MT-B-59-BMD-13-222-G) to W.S., and W.J. and Thailand Science Research and Innovation Fundamental Fund (Project no. 2323444) to W.S. and W.J.
Acknowledgements. We are grateful to Pisut Rimrang for assistance with flow cytometry experiments and analyses. The study was supported by Thammasat University Research Unit in Mineralized Tissue Reconstruction.

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
