## [Peer Review File · Royal Society Open Science]

Review History

RSOS-202066.R0 (Original submission)

Review form: Reviewer 1

Is the manuscript scientifically sound in its present form?

No

Are the interpretations and conclusions justified by the results?

No

Is the language acceptable?

Yes

Do you have any ethical concerns with this paper?

Yes

Have you any concerns about statistical analyses in this paper?

Yes

Recommendation?

Major revision is needed (please make suggestions in comments)

Comments to the Author(s)

Long term use of Zoledronic acid (ZA), an antiresorptive drug that reduces the isoprenoid intermediate and suppresses protein geranylation, increases the risk of MRONJ. Although geranylgeraniol (GGOH) is known to have an effect against ZA, exact mechanism is not clear. In this study, Singhatanadgit et al. showed that GGOH rescued the ZA-induced mineralization of mesenchymal stem cells (MSCs). Furthermore, they found that GGOH rescued the ZA-induced cell cycle arrest and apoptosis, as well as the RhoA activity and YAP. Although the rescue of ZA cytotoxic effects by GGOH-RhoA axis has been reported in other cells, the effects of GGOH on MSCs through RhoA and YAP pathway are potentially interesting. However, this referee has some concerns. Especially, the authors should strength the evidences for the involvement of YAP in this phenomenon.

Specific points

1. The evidence for YAP involvement is insufficient. LPA or DH are not a specific activator or inhibitor for YAP. These chemical target GPCR and have various function. Thus, authors need to examine the involvement of YAP using specific regulator, such as constitutive active YAP mutants and siRNA.
2. Fig. 3C. The width for S phase seems to be different in each condition. Same ranges should be used for each condition.
3. Fig. 4B. The author should explain the detailed methods for quantifying nuclear YAP in Method section.
4. Fig. 4B and 4C. The value for control, ZA, and ZA+GGOH seem to be identical between Fig 4B and Fig 4C. Are these experiments independent?
5. Fig. 4C, 4E, and 4G. Controls do not have error bars.
6. Page 7, line 21, "6 replicates" means biological replicates? Or just replication of PCR reaction? Biological replication is required.

Minor points

1. In Fig.1C, indicate the concentration of ZA.
2. In Fig.1F, what do the percent values (e.g. 15 +/- 3.8%) means?
3. Fig. 2B. Open bars are not visible in the print out.
4. Page 11, line 19. This reviewer does not understand how "88%" was calculated. Please explain it.
5. Page 13, Line 16-18. Please use same methods to indicate the decrease of cell proportion in each condition.
6. Page 29, line 10, (C)(D) should be (D)(E).

Review form: Reviewer 2

Is the manuscript scientifically sound in its present form?

Yes

Are the interpretations and conclusions justified by the results?

Yes

Is the language acceptable?

No

Do you have any ethical concerns with this paper?

No

Have you any concerns about statistical analyses in this paper?

No

Recommendation?

Accept with minor revision (please list in comments)

Comments to the Author(s)

See attached file (Appendix A).

Review form: Reviewer 3

Is the manuscript scientifically sound in its present form?

Yes

Are the interpretations and conclusions justified by the results?

Yes

Is the language acceptable?

Yes

Do you have any ethical concerns with this paper?

No

Have you any concerns about statistical analyses in this paper?

No

Recommendation?

Accept with minor revision (please list in comments)

Comments to the Author(s)

See attached file (Appendix B).

Decision letter (RSOS-202066.R0)

Dear Dr Singhatanadgit

The Editors assigned to your paper RSOS-202066 "Geranylgeraniol prevents zoledronic acid-mediated reduction of viable mesenchymal stem cells via induction of RhoA-dependent YAP activation" have now received comments from reviewers and would like you to revise the paper

in accordance with the reviewer comments and any comments from the Editors. Please note this decision does not guarantee eventual acceptance.

Please submit your revised manuscript and required files (see below) no later than 21 days from today's (ie 11-Jan-2021) date. Note: the ScholarOne system will 'lock' if submission of the revision is attempted 21 or more days after the deadline. If you do not think you will be able to meet this deadline please contact the editorial office immediately.

on behalf of Dr Simon Cook (Associate Editor) and Catrin Pritchard (Subject Editor)
openscience@royalsociety.org

Associate Editor Comments to Author (Dr Simon Cook):

Associate Editor: 1

Comments to the Author:

GGOH is shown to rescue the ZA induced reduction in MSC viability and ability of osteoclast-differentiated MSCs to induce mineralisation. Nuclear translocation of YAP is blocked by ZA and restored by GGOH and commensurate with this RhoA activity (GTP loading) (which can control YAP nuclear localisation) is inhibited by ZA and restored by GGOH. The authors go on to suggest that the protective effect GGOH is mediated by p21 and cyclin D1.

The manuscript describes some interesting results which are in principle acceptable for publication and the other referees seem to agree. But, as it stand, the study lacks critical controls, is too preliminary and over-interprets data. Major revision is required

In addition to the referees points the following key concerns must be addressed.

1. Experiments involving ZA + GGOH are 'drug combination experiments'. It is absolutely critical that all key observations are adequately controlled by using each drug in isolation as well as in combination. So Figure 1F, Figure 2A & 2B and Figure 3A-E, Figure 4-C must be repeated with ZA, GGOH and ZA+GGOH to firmly establish the 'combination effect' is real. All other results are dependent on this combination effect.
2. In Figure 4E the authors conclude that 'Increased YAP activation by LPA also restored the viability of cells treated with ZA'. LPA activates multiple effector pathways (ERK, PI3K, inhibition of Adenylyl Cyclase, etc). There is no evidence that the modest protective effect of LPA is due to YAP. Either perform YAP RNAi or remove this conclusion
3. The authors suggestion that the protective effect GGOH is mediated by p21 and cyclin D1 is not supported by their data. There is no evidence presented that GGOH regulates p21 or Cyc D1 in their system. They cannot extrapolate from other studies; they need to blot for p21 and Cyc D1. In addition, the compounds they use to address the role of p21 and Cyc D1 are very poorly characterised. For p21 UC2288 results need to be confirmed by p21 RNAi. For Cyc D1, Fascaplysin is actually a poorly characterised CDK4 inhibitor; so they are actually testing a CDK4 hypothesis. Cyc D1 does other things in addition to activating CDK4/6 so which hypothesis are they really testing? Cyc D1? Or CDK4. If Cyc D1 then repeat with RNAi. If CDK4 then repeat with clinically approved CDK4 inhibitors such as Palbociclib.

Reviewer comments to Author:

Reviewer: 1

Comments to the Author(s)

Long term use of Zoledronic acid (ZA), an antiresorptive drug that reduces the isoprenoid intermediate and suppresses protein geranylation, increases the risk of MRONJ. Although geranylgeraniol (GGOH) is known to have an effect against ZA, exact mechanism is not clear. In this study, Singhatanadgit et al. showed that GGOH rescued the ZA-induced mineralization of mesenchymal stem cells (MSCs). Furthermore, they found that GGOH rescued the ZA-induced cell cycle arrest and apoptosis, as well as the RhoA activity and YAP. Although the rescue of ZA cytotoxic effects by GGOH-RhoA axis has been reported in other cells, the effects of GGOH on MSCs through RhoA and YAP pathway are potentially interesting. However, this referee has some concerns. Especially, the authors should strengthen the evidences for the involvement of YAP in this phenomenon.

Specific points

1. The evidence for YAP involvement is insufficient. LPA or DH are not a specific activator or inhibitor for YAP. These chemical target GPCR and have various function. Thus, authors need to examine the involvement of YAP using specific regulator, such as constitutive active YAP mutants and siRNA.
2. Fig. 3C. The width for S phase seems to be different in each condition. Same ranges should be used for each condition.
3. Fig. 4B. The author should explain the detailed methods for quantifying nuclear YAP in Method section.
4. Fig. 4B and 4C. The value for control, ZA, and ZA+GGOH seem to be identical between Fig 4B and Fig 4C. Are these experiments independent?
5. Fig. 4C, 4E, and 4G. Controls do not have error bars.
6. Page 7, line 21, "6 replicates" means biological replicates? Or just replication of PCR reaction? Biological replication is required.

Minor points

1. In Fig.1C, indicate the concentration of ZA.
2. In Fig.1F, what do the percent values (e.g. 15 +/- 3.8%) means?
3. Fig. 2B. Open bars are not visible in the print out.

4. Page 11, line 19. This reviewer does not understand how “88%” was calculated. Please explain it.
5. Page 13, Line 16-18. Please use same methods to indicate the decrease of cell proportion in each condition.
6. Page 29, line 10, (C)(D) should be (D)(E).

Reviewer: 2

Comments to the Author(s)
See attached file

Reviewer: 3

Comments to the Author(s)
see attached file

===PREPARING YOUR MANUSCRIPT===

===PREPARING YOUR REVISION IN SCHOLARONE===

Author's Response to Decision Letter for (RSOS-202066.R0)

See Appendix C.

RSOS-202066.R1 (Revision)

Review form: Reviewer 1

Is the manuscript scientifically sound in its present form?

Yes

Are the interpretations and conclusions justified by the results?

Yes

Is the language acceptable?

Yes

Do you have any ethical concerns with this paper?

No

Have you any concerns about statistical analyses in this paper?

Yes

Recommendation?

Accept with minor revision (please list in comments)

Comments to the Author(s)

In this revised manuscript, some concerns were successfully addressed, but others are still remained.

1. In response to this reviewer's previous comment #3, the authors added the methods for quantifying nuclear YAP in Section 2.7 as "... Nuclear localization of YAP was determined by the presence of co-localization of YAP and DAPI...". However, this does not explain the criteria for "the presence of YAP". Do the authors determine "the presence" by eyes or the intensity ratio of nuclear localization to cytosolic localization? Ratios are commonly used.

2. The authors says that "These have been replaced by the correct values." in response to the comment #4, This reviewer just would like to confirm that these data are derived from independent raw data (because values for Fig4B and Fig5B are still very similar). If these data are derived from independent raw data, it is fine. If these data are from the same raw data, authors should combine two graphs to one.

3. Response to this reviewer's comment #5. The authors should normalize each value after combining triplicate experiments, then the authors can take the variation of control data into account.

Minor points

1. What do the values in Fig. 1F mean? Those values are the data of MTT assay or staining area of crystal violet? The authors should describe clearly.

2. Line 16 in Page 18 of the manuscript with change history. The authors describe “suggesting that other signaling pathways are involved in the protective effect of GGOH.” It would be better to describe that “YAP is involved in protective effect of GGOH” to summarize the importance of YAP, instead of (or in addition to) other signaling pathways.

Review form: Reviewer 2

Is the manuscript scientifically sound in its present form?

Yes

Are the interpretations and conclusions justified by the results?

Yes

Is the language acceptable?

Yes

Do you have any ethical concerns with this paper?

No

Have you any concerns about statistical analyses in this paper?

No

Recommendation?

Accept with minor revision (please list in comments)

Comments to the Author(s)

See attached file (Appendix D).

Decision letter (RSOS-202066.R1)

Dear Dr Singhatanadgit

On behalf of the Editors, we are pleased to inform you that your Manuscript RSOS-202066.R1 "Geranylgeraniol prevents zoledronic acid-mediated reduction of viable mesenchymal stem cells via induction of Rho-dependent YAP activation" has been accepted for publication in Royal Society Open Science subject to minor revision in accordance with the referees' reports. Please find the referees' comments along with any feedback from the Editors below my signature.

Please submit your revised manuscript and required files (see below) no later than 7 days from today's (ie 15-Apr-2021) date. Note: the ScholarOne system will 'lock' if submission of the revision is attempted 7 or more days after the deadline. If you do not think you will be able to meet this deadline please contact the editorial office immediately.

on behalf of Dr Simon Cook (Associate Editor) and Catrin Pritchard (Subject Editor)
openscience@royalsociety.org

Associate Editor Comments to Author (Dr Simon Cook):

Comments to the Author:

Your revised manuscript has now been reviewed again by two referees. You will see that they are happy overall with the revisions and recommend acceptance of the manuscript. However, you will also see that they have asked for some minor revisions. These are trivial and include text changes and statistical analysis. I must emphasize that these final revisions need to be addressed before your manuscript can be accepted.

Reviewer comments to Author:

Reviewer: 1

Comments to the Author(s)

In this revised manuscript, some concerns were successfully addressed, but others are still remained.

1. In response to this reviewer's previous comment #3, the authors added the methods for quantifying nuclear YAP in Section 2.7 as "... Nuclear localization of YAP was determined by the presence of co-localization of YAP and DAPI...". However, this does not explain the criteria for "the presence of YAP". Do the authors determine "the presence" by eyes or the intensity ratio of nuclear localization to cytosolic localization? Ratios are commonly used.

2. The authors says that "These have been replaced by the correct values." in response to the comment #4, This reviewer just would like to confirm that these data are derived from independent raw data (because values for Fig4B and Fig5B are still very similar). If these data are derived from independent raw data, it is fine. If these data are from the same raw data, authors should combine two graphs to one.

3. Response to this reviewer's comment #5. The authors should normalize each value after combining triplicate experiments, then the authors can take the variation of control data into account.

Minor points

1. What do the values in Fig. 1F mean? Those values are the data of MTT assay or staining area of crystal violet? The authors should describe clearly.
2. Line 16 in Page 18 of the manuscript with change history. The authors describe “suggesting that other signaling pathways are involved in the protective effect of GGOH.” It would be better to describe that “YAP is involved in protective effect of GGOH” to summarize the importance of YAP, instead of (or in addition to) other signaling pathways.

Reviewer: 2

Comments to the Author(s)

See attached file

===PREPARING YOUR MANUSCRIPT===

===PREPARING YOUR REVISION IN SCHOLARONE===

Author's Response to Decision Letter for (RSOS-202066.R1)

See Appendix E.

Decision letter (RSOS-202066.R2)

Dear Dr Singhatanadgit,

It is a pleasure to accept your manuscript entitled "Geranylgeraniol prevents zoledronic acid-mediated reduction of viable mesenchymal stem cells via induction of Rho-dependent YAP activation" in its current form for publication in Royal Society Open Science.

on behalf of Dr Simon Cook (Associate Editor) and Catrin Pritchard (Subject Editor)
openscience@royalsociety.org

Appendix A

Comments for the authors:

Geranylgeraniol prevents zoledronic acid-mediated reduction of viable mesenchymal stem cells via induction of RhoA-dependent YAP activation

Weerachai Singhatanadgit, Weerawan Hankamolsiri and Wanida Janvikul

In this manuscript the authors report on findings that the diterpene alcohol, geranylgeraniol, prevents zoledronic acid-induced reduction of MSCs via RhoA dependent YAP activation.

Prolonged treatment with ZA, to protect from osteoporosis, can result in osteonecrosis of the jaw, which has been attributed to ZA-mediated reduction in MSCs. ZA works by inhibiting protein geranylgeranylation, to suppress cell proliferation. Geranylgeraniol is an intermediate in the mevalonate pathway that can mitigate the effects of ZA; the mechanisms of this action are unknown.

Work presented in this manuscript showed a decrease the numbers of viable MSCs on exposure to ZA and this was reversed by GGOH: Ki67⁺ cell numbers, cell cycle arrest and apoptosis were all reversed by GGOH. The authors show GGOH reversed ZA-suppression of RhoA activity and YAP activation. Furthermore, they showed the involvement of YAP, cyclin D and p21 in the protective effects of GGOH. They suggest that GGOH may have promise as a therapeutic to prevent osteonecrosis of the jaw and that YAP could be a potential therapeutic target.

This manuscript presents interesting data but leaves many questions unanswered. It is possible that GGOH may be useful therapeutically but the problems likely to be encountered are underestimated here. The data are quite preliminary but are convincing in so far as they go. Some general revision of the grammar and language in the manuscript will be imperative before publication.

General points that should be included:

1. Hundreds of cellular proteins are geranylgeranylated generally making GGTase inhibitors, for example, toxic. The authors should comment on this.
2. Does YAP control expression of the genes investigated in section 3.2?
3. Rhosin also inhibits RhoB and RhoC. The authors should comment on this.
4. The authors should look at the phosphorylation levels of YAP in the presence of GGOH. A western blot would suffice.
5. If, as the authors suggest, GGOH directly inhibits the action of ZA, it may well interfere with the original therapeutic effects of ZA. This is a major concern regarding this work and although the authors allude to the issue it needs to be discussed in detail. Is there a therapeutic window for both ZA and GGOH to be administered successfully together? The authors suggest that local administration may be required, indicating that they do know this is an issue.

6. The discussion ends with a consideration of the roles of p21 in cell viability, however they make no conclusions. The authors should state what they think their own results actually mean.

Specific Corrections

p3, line 38: 'GGOH preserves a pool of viable MSCs with osteogenic potency against ZA by surrogating the activity of RhoA-dependent YAP activation' – surrogate is not a verb, rewrite

p4, line 26: 'with patients receiving more than 3 years of intravenous N-BPs possessing an increased risk of developing MRONJ' – what is the increase risk? please add figures

p4, line 38: What were the 'low' and 'high' doses and how do they relate to the clinical dose

p5, line 35: What are 'angiogenesis theory' and 'local toxicity theory'? Do the authors mean 'therapy'?

p6, line 14: 'is required for activation of Rho GTPases'. Change to 'is required for activity of Rho GTPases'

p12, line 24: 'and even much less viable cells.' Change to 'and far fewer viable cells'.

p12, line 26: change cytotoxic to cytotoxicity

p12, line 45: the authors suggest that data in Fig 1D show that ZA treatment decreased viable cell numbers to 88%. This seems to 65% in the figure.

p12, line 49-54: As the effects the authors are discussing are not significant, I would suggest this should be omitted.

p13, line 47: change 'survived' to rescued

p13, line 52: change 'survived' to rescued

p14, line 14 onwards: it is difficult to follow the beginning of section 3.3 What is the 'early event'? A sentence describing what the hypothesis is here and how they are investigating it is necessary.

p14, line 28: 'show a narrower distribution of cells in S phase' – suggest change to 'show a decrease in cells in S phase'

p14, line 47: 'In addition, more than 7 folds of apoptosis was observed' – poor grammar, rewrite. Additionally, how was this calculated? There are no details in M+Ms

p15, line 8: Again, a statement of the hypothesis being investigated is required to introduce section 3.4 (even if the involvement of YAP is explained in the introduction)

p15, lines 19 and 24: change nuYAP to nuclear YAP

p15, line 22: staining in the presence of ZA (Fig. 4B)

p15, line 34: 'geranylgeranylated protein, and thus activating the protein activities' change to 'geranylgeranylated protein levels, and thus activates Rho GTPase activity

p15, line 40: Do any other small G proteins regulate YAP?

p16, line 6: Change to 'A chemical inhibitor and an activator'

p16, line 8: Change 'used to link its role' to 'used to confirm its role'

p16, line 21: 'suggesting other signaling pathways are involved in the protective effect of GGOH.'

p16, line 36: We, therefore, used ~~the~~ chemical inhibitors targeting 2 key cell cycle regulators, cyclin D (fascaplysin) and p21 (UC2288), to determine their involvement in the protective effect of GGOH on ZA-suppressed cell viability.

p17, line 45-47: 'Nevertheless, in cancer patients, systemic infusion of MSCs may promote cancer metastasis and recurrence [40, 41].' Although this is true, are cancer patients a particularly relevant target group? Could this strategy just not be applicable unfortunately to cancer patients?

p18, lines 19-38: this paragraph is very useful to the reader and highly pertinent to the impact of the manuscript however it should be edited for further clarity as the poor grammar makes it difficult to understand.

p18, lines 3: change 5-10 uM to 5-10 μ M

p18, lines 41-43: The authors state that GGOH is unable to completely prevent ZA toxicity however in line 33 they state that 5-10 μ M GGOH 'effectively prevented' ZA-induced toxicity - please clarify.

p18, final paragraph: the authors proffer a reasonable hypothesis as to why GGOH does not completely rescue ZA-induced cytotoxicity (which could/should be tested in fact). How does Apppl induce cytotoxicity and why would this allow rescued MSCs to be clinically important (p19, line 3)?

p18, lines 54: However, the rescued MSCs ~~remained~~ retained their osteogenic function by forming bone-like mineralization, ~~under a proper induction~~ when induced, comparable to cells naïve to ZA, suggesting a clinical importance of these GGOH-protected MSCs.

p19, line 13: It has been shown that geranylgeranylation downstream of the mevalonate pathway is required and is sufficient to activate YAP and ~~subsequently result in activate ERK activation~~, which thus stimulates cell proliferation [47]

p19, line 36: Inhibition of cyclin D and p21 ~~also completely~~ diminished the cytoprotective effect of GGOH, suggesting that this effect was mediated through both cyclin D and p21.

p19, line 50: It has been suggested that the role of p21 may be dependent on its ~~localization, and dual behavior of p21 is greatly dependent on its~~ subcellular localization: Nuclear p21 acts mainly as a proliferation suppressor

p20, line 3: The p21 inhibitor UC2288

p20, line 26: 'adjuvant' is not the correct term here, as GGOH would be administered to counteract side-effects of ZA.

p30, line 35: and GGOH-protected MSCs ~~survived from ZA~~ were subsequently induced in osteogenic medium

p30, line 45: The biomineralization ~~experiments were repeated 3 times with similar results.~~ data shown are representative of three independent experiments (assuming that they were independent).

p33, Fig 1: Add the dose of ZA (5 μ M?) to each panel in (C) for clarity

p33, Fig 1: Add the dose of ZA (50 μ M?) and GGOH (10 μ M?) to each panel in (F) for clarity

p35, Fig 3: Add the dose of ZA (50 μ M?) and GGOH (10 μ M?) to each panel in (A-E) for clarity

p35, Fig 3 legend – (E) is missing

p36, Fig 4: Add the dose of ZA and GGOH throughout for clarity

p37, Fig 5: GGPPP does not 'activate RhoA- it is just a prerequisite that most small G proteins are lipid modified in order that they are correctly localized and can therefore be activated by a GEF – please edit figure appropriately.

Appendix B

Zoledronic acid (ZA), an inhibitor of protein geranyl-geranylation, is used in the clinic to treat osteoporosis and related bone diseases. However, long term use can contribute to osteonecrosis, linked to a decrease in mesenchymal stem cell (MSC) viability.

This study investigated the ability of geranyl-geraniol (GGOH) to reverse the detrimental effects of ZA on MSCs. GGOH counteracted the ZA induced reduction in MSC viability and ability of osteoclast-differentiated MSCs to induce mineralisation.

Nuclear translocation of the transcriptional activator Yap is shown to be blocked by ZA and restored by GGOH. Yap nuclear translocation can be stimulated by RhoA, a geranyl-geranylated Small GTPase. RhoA activity (GTP loading) is shown to be inhibited by ZA and restored by GGOH. The RhoA inhibitor Rhosin was used to implicate RhoA in the ZA/GGOH effects on Yap nuclear translocation.

The study is well performed, the data are convincing, well-controlled, well-analysed, and the paper is well-written. There are a few over-interpretations, mainly from the use of pharmacological reagents that have pleiotropic effects. These must be toned down. And some points need to be discussed.

- 1) I'm missing in Figs 1 and 4 the minus-ZA, plus-GGOH controls. I assume these have been done in earlier studies. If not, include here. If yes, please elaborate in the Results text on GGOH-only effects.
- 2) Lysophosphatidic acid is used as an activator and dobutamine hydrochloride as an inhibitor of Yap. These reagents activate GPCRs, and therefore have pleiotropic effects many of which are not mediated by Yap. The text must be toned down accordingly.
- 3) The identification of the RhoA-Yap axis is a major finding of this paper. It is done through the use of Rhosin, a small-molecule compound which inhibits the binding of several Rho-GEFs to RhoA. The authors should ideally confirm these Rhosin data through the use of a second pharmacological inhibitor of the RhoA pathway, or through interference with RhoA activity in another manner (e.g. expression of Rho-GEF, GAP, active/inactive RhoA proteins), or at least they should discuss these options.
- 4) Ref 20 shows that some effects of ZA are independent of RhoA, which is to be expected as RhoA is not the only geranyl-geranylated protein. Many other Small GTPases and other types of proteins are also geranyl-geranylated. Please elaborate in the discussion on other possible mediators of the ZA/GGOH effects.
- 5) The dose response in Fig 1 shows too little GGOH is ineffective, too much is cytotoxic. Please elaborate in the discussion how local dosage might be preventable in vivo.
- 6) Yap is not the only target of RhoA and can be activated in RhoA-independent manner. Please elaborate in the discussion on other possible downstream effectors of RhoA and upstream regulators of Yap in this context.

Typos

P5, line 36: therapy not theory?

P12, line 24: fewer not less viable cells?

Appendix C

Note to Editor

We greatly appreciate the constructive and thoroughly detailed comments from the Associate Editor and all the reviewers for We now include our point-by-point responses to each of the comments, and key/significant revised points are summarized as follows:

1. Additional new experiments were performed, as suggested, and the results are shown in revised Figures 4C-4D (Phosphorylation of YAP), 5D-5F (YAP siRNA efficiency and its role in the viability of GGOH/ZA-treated cells), 6A-6F (CDK4/6 activity and YAP-mediated CDK6). Discussion regarding these results are added.
2. The results of the repeated experiments that include the GGOH control group are shown in revised Figures 1F, 3A-3E, 2A-2B and 4A-4B. Additional comments on the lack of GGOH effects in the absence of ZA are included.
3. To accurately present the key finding, the previously claimed 'RhoA' involvement is toned down and changed to 'Rho' involvement throughout the revised manuscript, including the Title.

Response to Comments

Associate Editor:

1. Experiments involving ZA + GGOH are 'drug combination experiments'. It is absolutely critical that all key observations are adequately controlled by using each drug in isolation as well as in combination. So Figure 1F, Figure 2A & 2B and Figure 3A-E, Figure 4-C must be repeated with ZA, GGOH and ZA+GGOH to firmly establish the 'combination effect' is real. All other results are dependent on this combination effect.

Response: Experiments were repeated with ZA, GGOH and ZA+GGOH, as suggested, and the results and representative figures have been included in the respective Sections and Figures (revised Figures 1F, 3A-3E, 2A-2B and 4A-4B). In addition, additional comments regarding the lack of effects of GGOH alone (without ZA) have been added in the Discussion (Pages 24-25).

2. In Figure 4E the authors conclude that 'Increased YAP activation by LPA also restored the viability of cells treated with ZA'. LPA activates multiple effector pathways (ERK, PI3K, inhibition of Adenylyl Cyclase, etc). There is no evidence that the modest protective effect of LPA is due to YAP. Either perform YAP RNAi or remove this conclusion

Response: Experiments using siRNA specific to YAP were carried out to confirm the role of YAP. The results and representative figures have been added in the Results Section 3.5 (Pages 17-18). Discussion and Figure legend have also been revised accordingly.

3. The authors suggestion that the protective effect GGOH is mediated by p21 and cyclin D1 is not supported by their data. There is no evidence presented that GGOH regulates p21 or Cyc D1 in their system. They cannot extrapolate from other studies; they need to blot fo p21 and Cyc D1. In addition, the compounds they use to address the role of p21 and Cyc D1 are very poorly characterised. For p21 UC2288 results need to be confirmed by p21 RNAi. For Cyc D1, Fascaplysin is actually a poorly characterised CDK4 inhibitor; so they are actually tresting a CDK4 hypothesis. Cyc D1 does other things in addition to activating CDK4/6 so which hypothesis are they really testing? Cyc D1? Or CDK4. If Cyc D1 then repeat with RNAi. If CDK4 then repeat with clinically approved CDK4 inhibitors such as Palbociclib.

Response: We totally agree with the comment and we therefore focus on the hypothesis about CDK4/6, which are possible downstream effectors of YAP. Therefore, experiments using palbociclib were performed as suggested by the Associate Editor, and the results have been included in the Results Section 3.6 (Pages 18-19) and Fig. 6. The Materials and Methods (page??), Discussion (page??) have been revised accordingly. Fig. 6 and the corresponding Figure legend have been added. Moreover, by focusing on the YAP-CDK axis, the original results related to Cyclin D1 and p21 have been removed and in-depth results of the involvement of CDK4/6 have been included. We really appreciate the suggestion and guidance of the Associate Editor.

Reviewer: 1

Specific points

1. The evidence for YAP involvement is insufficient. LPA or DH are not a specific activator or inhibitor for YAP. These chemical target GPCR and have various function. Thus, authors need to examine the involvement of YAP using specific regulator, such as constitutive active YAP mutants and siRNA.

Response: Experiments using siRNA specific to YAP were carried out to confirm the role of YAP. The results and representative figures have been added in the Results Section 3.5 (Pages 17-18). Discussion and Figure legend have also been revised accordingly.

2. Fig. 3C. The width for S phase seems to be different in each condition. Same ranges should be used for each condition.

Response: The results have been re-evaluated and shown in Fig. 3C and 3D, as suggested.

3. Fig. 4B. The author should explain the detailed methods for quantifying nuclear YAP in Method section.

Response: The following details have been added in Section 2.7 (Page 10), as suggested.

“.....The NIS-Elements software was used for the image analysis, with Z-stacks of images being acquired for each channel. The middle confocal slice was chosen from the images at the middle nuclear plane, detected in the DAPI channel, and on the same slice, the image was acquired in the YAP channel to determine the localization of YAP. Nuclear localization of YAP was determined by the presence of co-localization of YAP and DAPI. The percentage of nuclear YAP-positive cells was measured from a total of 100 cells.“

4. Fig. 4B and 4C. The value for control, ZA, and ZA+GGOH seem to be identical between Fig 4B and Fig 4C. Are these experiments independent?

Response: These have been replaced by the correct values.

5. Fig. 4C, 4E, and 4G. Controls do not have error bars.

Response: As the values of the control group were adjusted to 1.0 or 100%. Therefore, the control groups do not have error bars.

6. Page 7, line 21, “6 replicates” means biological replicates? Or just replication of PCR reaction? Biological replication is required.

Response: Detailed information has been added in the Materials and Methods (Page 8) for clarity, as suggested.

Minor points

1. In Fig.1C, indicate the concentration of ZA.

Response: This has been revised, as suggested.

2. In Fig.1F, what do the percent values (e.g. 15 +/- 3.8%) means?

Response: This means 15 +/- 3.8% of the control defined as 100%.

Fig. 2B. Open bars are not visible in the print out.

Response: This has been revised, as suggested.

4. Page 11, line 19. This reviewer does not understand how “88%” was calculated. Please explain it.

Response: The value “88%” was mistakenly calculated, and the correct value has been added.

5. Page 13, Line 16-18. Please use same methods to indicate the decrease of cell proportion in each condition.

Response: This has been revised, as suggested, and the results are shown in Section 3.3.

6. Page 29, line 10, (C)(D) should be (D)(E).

Response: This has been revised, as suggested.

Reviewer 2

1. Hundreds of cellular proteins are geranylgeranylated generally making GGTase inhibitors, for example, toxic. The authors should comment on this.

Response: The following comment has been added in the Discussion (Page 25), as suggested.

“Due to the ubiquitous expression of a majority of Rho GTPases and their implication in fundamental cellular processes, their endogenous expressions and activities may be tightly regulated to ensure that cell division progresses properly without tumor transformation. Dysregulation of Rho GTPase signaling is closely associated with tumorigenesis and malignant phenotypes [60]. It has been shown that protein isoprenylation-dependent mechanism governing Rho GTPases turnover may represent a mechanism by which cells maintain a proper level of Rho-dependent cell signaling [61-63]. Moreover, some less understood posttranslational modifications, such as sumoylation, ubiquitylation, AMPylation, and transglutamination, may also regulate the activity of Rho GTPases [64]. In the present study, exogenously supplemented GGOH alone did not increase the level of endogenous Rho activity, YAP activation, CDK6, and the number of viable MSCs. However, when the activity of Rho was suppressed by ZA, GGOH stimulation of Rho activity, YAP activation, CDK6 and the number of viable MSCs became evident. The precise mechanism(s) underlying this remain(s) to be investigated. It is possible that negative feedback control could occur in response to the increased basal level of endogenous geranylgeranylation by the GGOH treatment alone. When the level of endogenous geranylgeranylation is reduced by ZA, exogenously added GGOH could help maintain the basal level of geranylgeranylation required for fundamental cellular processes, such as cell cycle progression. Moreover, a narrow range of GGOH concentrations that are optimal for cytoprotection against ZA observed in the present study might be attributed to a number of highly regulated mechanisms for controlling Rho GTPase-mediated cellular functions in MSCs. This could be responsible for the prevention of uncontrolled cell cycle progression and cell proliferation by the exogenously added GGOH. Careful in vivo determination of an effective dose level is required to obtain its accurate dose response relationship. The in vitro information will help support its safety for future animal and clinical studies.”

2. Does YAP control expression of the genes investigated in section 3.2?

Response: We did not look at the effect of YAP on the expression of these osteogenic genes, but previous studies reported conflicting results (Kovar et al. 2020). These discrepancies may be due to stage of differentiation of the stem cells towards osteoblasts, and this is also of our interest. Further work will be carried out to prove this hypothesis.

(Kovar H, Bierbaumer L, Radic-Sarikas B. 2020. The yap/taz pathway in osteogenesis and bone sarcoma pathogenesis. *Cells*. 9(4):972.)

3. Rhosin also inhibits RhoB and RhoC. The authors should comment on this.

Response: The following comment has been added in the Discussion (Pages 21-22), as suggested.

“The present results demonstrated the involvement of Rho in the cytoprotection of GGOH against ZA and suggested that among the 3 isoforms in the Rho subfamily, RhoA could play a role as its reduced activity caused by ZA was reversed by GGOH. However, we cannot rule out the possibility that RhoB and RhoC may also be involved in the reversal effect of GGOH. Rhosin, a small-molecule compound used in the present study, inhibits the activity of RhoA, RhoB and RhoC. The use of a second pharmacological inhibitor of the RhoA pathway or interference with RhoA activity in another manner (e.g., targeting the expression of guanine nucleotide exchange factors (GEFs), GTPase-activating proteins (GAPs), guanine-

dissociation inhibitors (GDI) and active/inactive RhoA) will undoubtedly establish the functional importance of RhoA specifically in GGOH-mediated cytoprotection against ZA in MSCs. It is also noteworthy that some effects of ZA may also be Rho-independent as Rho proteins are not the only geranylgeranylated proteins in MSCs. Ras, Cdc42 and Rac GTPases and other types of proteins, such as heterotrimeric G proteins, are also geranylgeranylated. Decreased prenylation of other small GTPases, such as Rap, Ras, and Cdc42, has been linked to some effects of ZA [40-43]. These proteins are essential for multiple cellular processes, including cell proliferation, cell movement, cytoskeletal rearrangement and apoptosis [19, 44, 45]. Whether or not these molecules are involved in the protective effect of GGOH on ZA-induced MRONJ requires further studies. In addition, the identification of the proteins that became unprenylated after ZA treatment should be of great interest in understanding the interaction of signaling pathways triggered by ZA and rescued by GGOH.”

4. The authors should look at the phosphorylation levels of YAP in the presence of GGOH. A western blot would suffice.

Response: The results have been added in Section 3.4 (Pages 16-17) and Fig. 4, as suggested.

5. If, as the authors suggest, GGOH directly inhibits the action of ZA, it may well interfere with the original therapeutic effects of ZA. This is a major concern regarding this work and although the authors allude to the issue it needs to be discussed in detail. Is there a therapeutic window for both ZA and GGOH to be administered successfully together? The authors suggest that local administration may be required, indicating that they do know this is an issue.

Response: The following comments have been added in the Discussion (Pages 20-21), as suggested.

“It is proposed that GGOH can be locally and directly administered using a biocompatible carrier into the necrotic bone defect, or the extraction socket, of the jaw bone to treat, or prevent, MRONJ. The local application of GGOH therapy is therefore beneficial especially in cancer patients as this method helps circumvent the aforementioned side effects (from systemic delivery of MSCs) as well as avoids GGOH interference of osteoclast-inhibiting clinical benefit of N-BPs that is required in other sites of skeletal bones. The presence of GGOH within a local area in the jaw bone will help preserve the viability of MSCs against the released unbound form of ZA during the bone repair without interfering the effect of ZA in other skeletal bones such as femur and hip. Diffusion of the locally administered GGOH to these distant bones is possible, but the concentration may not be high enough to elicit any effect against ZA.”

6. The discussion ends with a consideration of the roles of p21 in cell viability, however they make no conclusions. The authors should state what they think their own results actually mean.

Response: The results regarding p21 have been removed, as a concern raised by the Associate Editor, and the discussion was thus removed.

Specific Corrections

p3, line 38: ‘GGOH preserves a pool of viable MSCs with osteogenic potency against ZA by surrogating the activity of RhoA-dependent YAP activation’ – surrogate is not a verb, rewrite

Response: This has been revised to “... by rescuing...”.

p4, line 26: ‘with patients receiving more than 3 years of intravenous N-BPs possessing an

increased risk of developing MRONJ' – what is the increase risk? please add figures

Response: This has been added in Page 3, as suggested.

The cumulative incidence of developing MRONJ among patients receiving intravenous N-BPs increased from 0.5-0.8% at 1 year to 1.3-4.3% at 3 years without a plateau after 2-3 years as reported for patients receiving an alternative antiresorptive drug denosumab [1, 3].

p4, line 38: What were the 'low' and 'high' doses and how do they relate to the clinical dose

Response: This has been revised, as suggested.

p5, line 35: What are 'angiogenesis theory' and local toxicity theory'? Do the authors mean 'therapy'?

Response: The word "theory" has been changed to "therapy" for clarity, as suggested.

p6, line 14: 'is required for activation of Rho GTPases'. Change to 'is required for activity of Rho GTPases'

Response: This has been revised, as suggested.

p12, line 24: 'and even much less viable cells.' Change to 'and far fewer viable cells'.

Response: This has been revised, as suggested.

p12, line 26: change cytotoxic to cytotoxicity

Response: This has been revised, as suggested.

p12, line 45: the authors suggest that data in Fig 1D show that ZA treatment decreased viable cell numbers to 88%. This seems to 65% in the figure.

Response: The value "88%" has been revised to the correct value "64%".

p12, line 49-54: As the effects the authors are discussing are not significant, I would suggest this should be omitted.

Response: This has been revised, as suggested.

p13, line 47: change 'survived' to rescued

Response: This has been revised, as suggested.

p13, line 52: change 'survived' to rescued

Response: This has been revised, as suggested.

p14, line 14 onwards: it is difficult to follow the beginning of section 3.3 What is the 'early event'? A sentence describing what the hypothesis is here and how they are investigating it is necessary.

Response: In this context, the early events include changes in the number of proliferating cells, cell cycle and apoptosis. This has been revised for clarity, and a statement describing what the hypothesis is and how it was investigated has been added, as suggested.

p14, line 28: 'show a narrower distribution of cells in S phase' – suggest change to 'show a decrease in cells in S phase'

Response: This has been revised, as suggested.

p14, line 47: 'In addition, more than 7 folds of apoptosis was observed' – poor grammar, rewrite. Additionally, how was this calculated? There are no details in M+Ms

Response: ‘In addition, more than 7 folds of apoptosis was observed’ has been changed to ‘In addition, a 7-fold increase in apoptosis was observed’. The calculation was emphasized in the Materials and Methods.

p15, line 8: Again, a statement of the hypothesis being investigated is required to introduce section 3.4 (even if the involvement of YAP is explained in the introduction)

Response: A statement “Since YAP activation, which is crucial for cell viability and proliferation, can be regulated by protein geranylgeranylation, it is possible that alteration of geranylgeranylation by ZA and GGOH may control the activation of YAP. Therefore, the effects of ZA and GGOH on nuclear localization and phosphorylation of YAP were examined.” To introduce Section 3.4 has been added, as suggested.

p15, lines 19 and 24: change nuYAP to nuclear YAP

Response: This has been revised, as suggested.

p15, line 22: staining in the presence of ZA (Fig. 4B)

Response: This has been revised, as suggested.

p15, line 34: ‘geranylgeranylated protein, and thus activating the protein activities’ change to ‘geranylgeranylated protein levels, and thus activates Rho GTPase activity

Response: This has been revised, as suggested.

p15, line 40: Do any other small G proteins regulate YAP?

Response: This has been discussed in page 21, as shown below.

“...It is also noteworthy that some effects of ZA may also be Rho-independent as Rho proteins are not the only geranylgeranylated proteins in MSCs. Ras, Cdc42 and Rac GTPases and other types of proteins, such as heterotrimeric G proteins, are also geranylgeranylated. Decreased prenylation of other small GTPases, such as Rap, Ras, and Cdc42, has been linked to some effects of ZA [41-44]. These proteins are essential for multiple cellular processes, including cell proliferation, cell movement, cytoskeletal rearrangement and apoptosis [20, 45, 46]. Whether or not these molecules are involved in the protective effect of GGOH on ZA-induced MRONJ requires further studies. In addition, the identification of the proteins that became unprenylated after ZA treatment should be of great interest in understanding the interaction of signaling pathways triggered by ZA and rescued by GGOH.”

p16, line 6: Change to ‘A chemical inhibitor and an activator’

Response: This has been revised, as suggested.

p16, line 8: Change ‘used to link its role’ to ‘used to confirm its role’

Response: This has been revised, as suggested.

p16, line 21: ‘suggesting other signaling pathways are involved in the protective effect of GGOH.’

Response: This has been revised, as suggested.

p16, line 36: We, therefore, used ~~the~~ chemical inhibitors targeting 2 key cell cycle regulators, cyclin D (fascaplysin) and p21 (UC2288), to determine their involvement in the protective effect of GGOH on ZA-suppressed cell viability.

Response: The results regarding p21 has been removed, as a concern raised by the Associate Editor, and the discussion was thus removed.

p17, line 45-47: ‘Nevertheless, in cancer patients, systemic infusion of MSCs may promote cancer metastasis and recurrence [40, 41].’ Although this is true, are cancer patients a particularly relevant target group? Could this strategy just not be applicable unfortunately to cancer patients?

Response: The following comments have been added in the Discussion (pages 20-21), as suggested.

“It is proposed that GGOH can be locally and directly administered using a biocompatible carrier into the necrotic bone defect, or the extraction socket, of the jaw bone to treat, or prevent, MRONJ. The local application of GGOH therapy is therefore beneficial especially in cancer patients as this method helps circumvent the aforementioned side effects (from systemic delivery of MSCs) as well as avoids GGOH interference of osteoclast-inhibiting clinical benefit of N-BPs that is required in other sites of skeletal bones. The presence of GGOH within a localized area in the jaw bone will help preserve the viability of MSCs against the released unbound form of ZA during the bone repair without interfering the effect of ZA in other skeletal bones such as femur and hip. Diffusion of the locally administered GGOH to these distant bones is possible, but its concentration may not be high enough to elicit any effects against the entire ZA in the body.”

p18, lines 19-38: this paragraph is very useful to the reader and highly pertinent to the impact of the manuscript however it should be edited for further clarity as the poor grammar makes it difficult to understand.

Response: This has been revised and shown in page 22 of the revised version, as suggested.

p18, lines 3: change 5-10 uM to 5-10 μ M

Response: This has been revised, as suggested.

p18, lines 41-43: The authors state that GGOH is unable to completely prevent ZA toxicity however in line 33 they state that 5-10 μ M GGOH ‘effectively prevented’ ZA-induced toxicity - please clarify.

Response: The phrase “effectively prevented ZA-induced toxicity” has been revised to “provided maximum protection from ZA-induced toxicity”.

p18, final paragraph: the authors proffer a reasonable hypothesis as to why GGOH does not completely rescue ZA-induced cytotoxicity (which could/should be tested in fact). How does ApppI induce cytotoxicity and why would this allow rescued MSCs to be clinically important (p19, line 3)?

Response: The following comment has been added in the discussion (Page 23).

“It is also important to note that ApppI inhibits the mitochondrial adenine nucleotide translocase (ANT), thus resulting in cell apoptosis [46]. The expression of ANT can be regulated by various intracellular signaling pathways initiated by growth factors and cytokines [47, 48]. It is therefore possible that during an initial phase of bone healing, a wide range of anabolic mediators involved in the proliferation and survival of stem/progenitor cells may modulate mitochondrial energy metabolism and the expression of ANT, and thus protecting MSCs from ApppI-induced cytotoxicity. Further studies are required to test this hypothesis and identify key molecules that effectively prevent ApppI-mediated cytotoxicity induced by ZA.”

p18, lines 54: However, the rescued MSCs ~~remained~~ retained their osteogenic function by forming bone-like mineralization, ~~under a proper induction~~ when induced, comparable to cells naïve to ZA, suggesting a clinical importance of these GGOH-protected MSCs.

Response: This has been revised, as suggested.

p19, line 13: It has been shown that geranylgeranylation downstream of the mevalonate pathway is required and is sufficient to activate YAP and ~~subsequently result in~~ activate ERK activation, which thus stimulates cell proliferation [47]

Response: This has been revised, as suggested.

p19, line 36: Inhibition of cyclin D and p21 ~~also completely~~ diminished the cytoprotective effect of GGOH, suggesting that this effect was mediated through both cyclin D and p21.

Response: The results regarding p21 has been removed, as a concern raised by the Associate Editor, and the discussion was thus removed.

p19, line 50: It has been suggested that the role of p21 may be dependent on its ~~localization, and dual behavior of p21 is greatly dependent on its~~ subcellular localization: Nuclear p21 acts mainly as a proliferation suppressor

Response: The results regarding p21 has been removed, as a concern raised by the Associate Editor, and the discussion was thus removed.

p20, line 3: The p21 inhibitor UC2288

Response: The results regarding p21 has been removed, as a concern raised by the Associate Editor, and the discussion was thus removed.

p20, line 26: ‘adjuvant’ is not the correct term here, as GGOH would be administered to counteract side-effects of ZA.

Response: The term ‘adjuvant’ has been changed to “...GGOH as an attractive pharmacological compound for...”.

p30, line 35: and GGOH-protected MSCs ~~survived from ZA~~ were subsequently induced in osteogenic medium

Response: This has been revised, as suggested.

p30, line 45: The biomineralization ~~experiments were repeated 3 times with similar results.~~ data shown are representative of three independent experiments (assuming that they were independent).

Response: This has been revised, as suggested.

p33, Fig 1: Add the dose of ZA (5 μ M?) to each panel in (C) for clarity

Response: This has been revised, as suggested.

p33, Fig 1: Add the dose of ZA (50 μ M?) and GGOH (10 μ M?) to each panel in (F) for clarity

Response: This has been revised, as suggested.

p35, Fig 3: Add the dose of ZA (50 μ M?) and GGOH (10 μ M?) to each panel in (A-E) for clarity

Response: This has been revised, as suggested.

p35, Fig 3 legend – (E) is missing

Response: This has been added, as suggested.

p36, Fig 4: Add the dose of ZA and GGOH throughout for clarity

Response: This has been revised, as suggested.

p37, Fig 5: GGPPP does not 'activate RhoA- it is just a prerequisite that most small G proteins are lipid modified in order that they are correctly localized and can therefore be activated by a GEF – please edit figure appropriately.

Response: The figure has been edited, as suggested. (Original Fig. 5 is now shown in Fig. 7 of the revised manuscript).

Reviewer 3

1) I'm missing in Figs 1 and 4 the minus-ZA, plus-GGOH controls. I assume these have been done in earlier studies. If not, include here. If yes, please elaborate in the Results text on GGOH-only effects.

Response: Experiments were repeated with ZA, GGOH and ZA+GGOH, as suggested, and the results and representative figures have been included in the respective Sections and revised Figures 1F, 3A-3E, 2A-2B and 4A-4B. In addition, additional comments regarding the lack of effects of GGOH alone (without ZA) have been added in the Discussion (pages 24-25).

2) Lysophosphatidic acid is used as an activator and dobutamine hydrochloride as an inhibitor of Yap. These reagents activate GPCRs, and therefore have pleiotropic effects many of which are not mediated by Yap. The text must be toned down accordingly.

Response: Experiments using siRNA specific to YAP were carried out to confirm the role of YAP. The results and representative figures have been added in the Results Section 3.5 (pages 17-18). Discussion and Figure legend have also been revised accordingly.

3) The identification of the RhoA-Yap axis is a major finding of this paper. It is done through the use of Rhosin, a small-molecule compound which inhibits the binding of several Rho-GEFs to RhoA. The authors should ideally confirm these Rhosin data through the use of a second pharmacological inhibitor of the RhoA pathway, or through interference with RhoA activity in another manner (e.g. expression of Rho-GEF, GAP, active/inactive RhoA proteins), or at least they should discuss these options.

Response: The following comment has been discussed, as suggested (Page 21).

“The present results demonstrated the involvement of Rho in the cytoprotection of GGOH against ZA and suggested that among the 3 isoforms in the Rho subfamily, RhoA could play a role as its reduced activity caused by ZA was reversed by GGOH. However, we cannot rule out the possibility that RhoB and RhoC may also be involved in the reversal effect of GGOH. Rhosin, a small-molecule compound used in the present study, inhibits the activity of RhoA, RhoB and RhoC. The use of a second pharmacological inhibitor of the RhoA pathway or interference with RhoA activity in another manner (e.g., targeting the expression of guanine nucleotide exchange factors (GEFs), GTPase-activating proteins (GAPs), guanine-dissociation inhibitors (GDI) and active/inactive RhoA) will undoubtedly establish the functional importance of RhoA specifically in GGOH-mediated cytoprotection against ZA in MSCs. It is also noteworthy that some effects of ZA may also be Rho-independent as Rho proteins are not the only geranylgeranylated proteins in MSCs. Ras, Cdc42 and Rac GTPases and other types of proteins, such as heterotrimeric G proteins, are also geranylgeranylated. Decreased prenylation of other small GTPases, such as Rap, Ras, and Cdc42, has been linked to some effects of ZA [40-43]. These proteins are essential for multiple cellular processes, including cell proliferation, cell movement, cytoskeletal rearrangement and apoptosis [19, 44, 45]. Whether or not these molecules are involved in the protective effect of GGOH on ZA-induced MRONJ requires further studies. In addition, the identification of the proteins that became unprenylated after ZA treatment should be of great interest in understanding the interaction of signaling pathways triggered by ZA and rescued by GGOH.”

4) Ref 20 shows that some effects of ZA are independent of RhoA, which is to be expected as RhoA is not the only geranyl-geranylated protein. Many other Small GTPases and other types of proteins are also geranyl-geranylated. Please elaborate in the discussion on other

possible mediators of the ZA/GGOH effects.

Response: This has been discussed, as suggested (Page 21).

5) The dose response in Fig 1 shows too little GGOH is ineffective, too much is cytotoxic. Please elaborate in the discussion how local dosage might be preventable in vivo.

Response: The narrow optimal/preventive concentration range has been discussed (shown below), as suggested (Page 25).

“...Moreover, a narrow range of GGOH concentrations that are optimal for cytoprotection against ZA observed in the present study might be attributed to a number of highly regulated mechanisms for controlling Rho GTPase-mediated cellular functions in MSCs. This could be responsible for the prevention of uncontrolled cell cycle progression and cell proliferation by the exogenously added GGOH. Careful in vivo determination of an effective dose level is required to obtain its accurate dose response relationship. The in vitro information will help support its safety for future animal and clinical studies.”

6) Yap is not the only target of RhoA and can be activated in RhoA-independent manner. Please elaborate in the discussion on other possible downstream effectors of RhoA and upstream regulators of Yap in this context.

Response: Discussion as shown below has been added in the discussion section, as suggested.

“However, it is also possible that in this context, other downstream effectors of RhoA can also mediate cell division and proliferation of MSC. These effectors include actin stress fiber, cyclin D1 and cyclin-dependent kinase inhibitors p21 and p27 [21, 36]. Moreover, RhoA-independent regulators of YAP, such as PDZ binding-kinase (PBK), IGF-1 and Wnt signaling [36, 63, 64], could also contribute to the cytoprotection of GGOH. Further studies are needed to advance our knowledge about the mechanisms controlling GGOH effect against ZA cytotoxicity.”

Typos

P5, line 36: therapy not theory?

Response: The word “theory” has been changed to “therapy” for clarity, as suggested.

P12, line 24: fewer not less viable cells?

Response: The phrase ‘and even much less viable cells.’ has been changed to ‘and far fewer viable cells’.

Appendix D

Comments for the authors:

Geranylgeraniol prevents zoledronic acid-mediated reduction of viable mesenchymal stem cells via induction of Rho-dependent YAP activation

Weerachai Singhatanadgit, Weerawan Hankamolsiri and Wanida Janvikul

This is a revised version of a manuscript submitted in December 2020. The manuscript is considerably improved and a substantial amount of new data is presented. Most of the concerns of this reviewer have been addressed satisfactorily.

The revised manuscript includes new data showing pYAP levels, siRNA knock down of YAP to demonstrate its role in GGOH-ZA treated cells and data indicating a role for CDK6 in the signalling under investigation. These data, and the new discussion on the pathways involved, strengths the manuscript appreciably and makes it suitable for publication.

There a few minor points that should be addressed. These often involve clarifying the difference between Rho family and Rho subfamily (members). I suggest using 'family' when talking about all 22 Rho family proteins and 'subfamily' to denote RhoA, B and C (when they cannot be distinguished).

Specific Corrections

p4, line 24: The Rho subfamily GTPases play an important role – if indeed this is what the authors mean.

p4, line 29: 'High Rho GTPase activity' – work in the reference cited here does indicate RhoA involvement.

p4, line 31: 'Inhibition of these small GTPases' -which ones?

p8, line 17: 'Each of the gene signals_s was normalized'

p17, line 28: 'including Rho GTPases' change to 'including Rho family GTPases'

p17, line 34: 'Among Rho GTPases a subfamily Rho plays and important role' change to 'The Rho subfamily itself (Rho A, B and C) plays an important role'.

p17, line 40: 'We confirmed the involvement of Rho in the cytoprotective effect' change to 'We confirmed the involvement of Rho subfamily proteins in the cytoprotective effect'

p17, line 45: 'results showed that the RhoA activity in cells treated' change to 'results showed that the RhoA·GTP levels in cells'

p17, line 49: 'Although GGOH did not alter the activity of RhoA, it significantly restored the level of RhoA activity to approximately' change to 'Although GGOH did not alter the levels of RhoA·GTP, it significantly restored the level of RhoA·GTP to approximately'

p17, line 54: 'Moreover, when the activity of Rho was blocked' change to 'Moreover, when the activity of Rho subfamily proteins was blocked'

p18, line 26-29: 'To unequivocally establish the role of YAP, ~~the~~ siRNA specific to YAP was used and ~~the~~ YAP protein expression was suppressed'

p22, line 6: 'Rho subfamily proteins and YAP have been shown to play important roles' - if this is what is shown in the references cited here.

p22, line 10: 'involvement of Rho subfamily proteins in the cytoprotecting'

p24, line 24: 'phosphorylated YAP ~~is-accumulated~~s in the cytoplasm'

Figure 5A: change y axis label to RhoA·GTP levels (%)

Figure 7: Change RhoA to Rho(A) in both places this is used.

Appendix E

Response to Reviewers' Comments

Reviewer: 1

1. In response to this reviewer's previous comment #3, the authors added the methods for quantifying nuclear YAP in Section 2.7 as "... Nuclear localization of YAP was determined by the presence of co-localization of YAP and DAPI...". However, this does not explain the criteria for "the presence of YAP". Do the authors determine "the presence" by eyes or the intensity ratio of nuclear localization to cytosolic localization? Ratios are commonly used.

Response: A sentence "Nuclear localization of YAP was determined by the presence of co-localization of YAP and DAPI with the intensity ratio of nuclear localization to cytosolic localization being higher than 1.0." has been added clarity.(Page 10)

2. The authors says that "These have been replaced by the correct values." in response to the comment #4, This reviewer just would like to confirm that these data are derived from independent raw data (because values for Fig4B and Fig5B are still very similar). If these data are derived from independent raw data, it is fine. If these data are from the same raw data, authors should combine two graphs to one.

Response: We confirm that these data are derived from independent raw data.

3. Response to this reviewer's comment #5. The authors should normalize each value after combining triplicate experiments, then the authors can take the variation of control data into account.

Response: We have recalculated as suggested by the reviewer and the standard deviations of the controls in all related Figures have now been included. (Figures 1A-B, 1D-F, 2B, 3A, 4D 5A, 5C, 5E-F6A, 6C-F)

Minor points

1. What do the values in Fig. 1F mean? Those values are the data of MTT assay or staining area of crystal violet? The authors should describe clearly.

Response: A sentence "The values in (F) are derived from MTT assay." Has been added in Figure legend 1 for clarity.

2. Line 16 in Page 18 of the manuscript with change history. The authors describe "suggesting that other signaling pathways are involved in the protective effect of GGOH." It would be better to describe that "YAP is involved in protective effect of GGOH" to summarize the importance of YAP, instead of (or in addition to) other signaling pathways.

Response: This has been changed as suggested.

Reviewer 2

There are a few minor points that should be addressed. These often involve clarifying the difference between Rho family and Rho subfamily (members). I suggest using 'family' when talking about all 22 Rho family proteins and 'subfamily' to denote RhoA, B and C (when they cannot be distinguished).

Response: This has been changed as suggested.

Specific Corrections

p4, line 24: The Rho subfamily GTPases play an important role – if indeed this is what the authors mean.

Response: This has been changed as suggested.

p4, line 29: 'High Rho GTPase activity' – work in the reference cited here does indicate RhoA involvement.

Response: 'High Rho GTPase activity' has been changed to 'High RhoA GTPase activity'

p4, line 31: 'Inhibition of these small GTPases' -which ones?

Response: 'Inhibition of these small GTPases' has been changed to 'Inhibition of these Rho subfamily GTPases'

p8, line 17: 'Each of the gene signals was normalized'

Response: This has been changed as suggested.

p17, line 28: 'including Rho GTPases' change to 'including Rho family GTPases'

Response: This has been changed as suggested.

p17, line 34: 'Among Rho GTPases a subfamily Rho plays an important role' change to 'The Rho subfamily itself (Rho A, B and C) plays an important role'.

Response: This has been changed as suggested.

p17, line 40: 'We confirmed the involvement of Rho in the cytoprotective effect' change to 'We confirmed the involvement of Rho subfamily proteins in the cytoprotective effect'

Response: This has been changed as suggested.

p17, line 45: 'results showed that the RhoA activity in cells treated' change to 'results showed that the RhoA·GTP levels in cells'

Response: This has been changed as suggested.

p17, line 49: 'Although GGOH did not alter the activity of RhoA, it significantly restored the level of RhoA activity to approximately' change to 'Although GGOH did not alter the levels of RhoA·GTP, it significantly restored the level of RhoA·GTP to approximately'

Response: This has been changed as suggested.

p17, line 54: 'Moreover, when the activity of Rho was blocked' change to 'Moreover, when the activity of Rho subfamily proteins was blocked'

Response: This has been changed as suggested.

p18, line 26-29: 'To unequivocally establish the role of YAP, ~~the~~ siRNA specific to YAP was used and ~~the~~ YAP protein expression was suppressed'

Response: This has been changed as suggested.

p22, line 6: 'Rho subfamily proteins and YAP have been shown to play important roles' - if this is what is shown in the references cited here.

Response: This has been changed as suggested.

p22, line 10: 'involvement of Rho subfamily proteins in the cytoprotecting'

Response: This has been changed as suggested.

p24, line 24: 'phosphorylated YAP is accumulated~~s~~ in the cytoplasm'

Response: This has been changed as suggested.

Figure 5A: change y axis label to RhoA·GTP levels (%)

Response: This has been changed as suggested.

Figure 7: Change RhoA to Rho(A) in both places this is used.

Response: This has been changed as suggested.